# Linear-Sample Learning of Low-Rank Distributions

Ayush Jain and Alon Orlitsky
Dept. of Electrical and Computer Engineering
University of California, San Diego
{ayjain, aorlitsky}@eng.ucsd.edu

## Abstract

Many latent-variable applications, including community detection, collaborative filtering, genomic analysis, and NLP, model data as generated by low-rank matrices. Yet despite considerable research, except for very special cases, the number of samples required to efficiently recover the underlying matrices has not been known.

We determine the onset of learning in several common latent-variable settings. For all of them, we show that learning $k \times k$, rank-$r$, matrices to normalized $L_1$ distance $\epsilon$ requires $\Omega(\frac{kr}{\epsilon^2})$ samples, and propose an algorithm that uses $\mathcal{O}(\frac{kr}{\epsilon^2} \log^2 \frac{r}{\epsilon})$ samples, a number linear in the high dimension, and nearly linear in the, typically low, rank.

The algorithm improves on existing spectral techniques and runs in polynomial time. The proofs establish new results on the rapid convergence of the spectral distance between the model and observation matrices, and may be of independent interest.

## 1 Introduction

### 1.1 Motivation

A great many scientific and technological applications concern relations between two objects that range over large domains, yet are linked via a low-dimensional *latent* space. Often this relation is precisely, or nearly, linear, hence can be modeled by a low-rank matrix. The problem of recovering low-rank model matrices from observations they generate has therefore been studied extensively. Following are five of the most common settings considered, each introduced via a typical application.

**Distribution matrices** In *Probabilistic latent semantic analysis* samples are co-occurrences $(w, d)$ of words and documents, assumed independent given one of $r$ latent topic classes $t$ [Hof99]. The joint probability matrix $[p_{w,d}]$ is therefore a mixture of at most $r$ product matrices $p(d|t) \cdot p(w|t)$, and has rank $\leq r$. This setting also arises in many hidden Markov applications, e.g., [MR05, HKZ12].

For this setting, the model matrix is a distribution, hence its elements are non-negative and sum to 1. The model is sampled independently $n$ times, and $X_{i,j}$ is the number of times pair $(i, j)$ was observed. In the remaining settings, the model matrix consists of arbitrary parameters, it is sampled just once, and each parameter is reflected in an independent observation $X_{i,j}$.

**Poisson Parameters** Recommendation systems infer consumer preferences from their consumption patterns. The number of times customer $i$ purchases product $j$ is typically modeled as an independent Poisson random variable $X_{i,j} \sim \text{Poi}(\lambda_{i,j})$. Often $\lambda_{i,j}$ is the inner product of the consumer's disposition and product's expression of $r$ latent features [SKKR00], so the parameter matrix $[\lambda_{i,j}]$ has rank $\leq r$.

**Bernoulli Parameters** In *inhomogeneous Erdös-Rényi graphs*, the edge between nodes $i$ and $j$ is an independent random variable $X_{i,j} \sim \text{Ber}(p_{i,j})$. In *Community-detection*, the Stochastic Block Model (SBM), *e.g.,* [Abb17, MNS18, ABH15, BLM15], assumes that graph nodes fall in few communities $C_1, \ldots, C_r$ such that the $p_{i,j}$'s are one constant if $i$ and $j$ are in the same community, and a different constant if $i$ and $j$ are in two distinct communities. Clearly, the parameter matrix $[p_{i,j}]$ has rank $\leq r$.

**Binomial Parameters** The probability that gene pair $(i, j)$ will express as a phenotype is often viewed as the result of a few factors, resulting in an expression probability matrix $[p_{i,j}]$ of low rank at most $r$ [KMA16]. In a study of $t$ phenotypic patients, the number $X_{i,j}$ of patients with gene pair $(i, j)$ will therefore be distributed binomially $\mathrm{Bin}(t, p_{i,j})$.

**Collaborative filtering** Let $Y_{i,j} \in [0, 1]$ be the random rating user $i$ assigns to movie $j$. If the ratings are based on a small number of $r$ intrinsic film features, the matrix $[F_{i,j}]$ of expected ratings $F_{i,j} = \mathbb{E}[Y_{i,j}]$, may be views as having a low rank $r$. Since only some ratings are reported, the matrix-completion and collaborative-filtering literature, e.g., [BCLS17], assumes that $\mathbb{1}_{i,j} \sim \mathrm{Ber}(p)$ are independent indicator random variables, and upon observing $X_{i,j} = \mathbb{1}_{i,j} \cdot Y_{i,j}$ for all $(i, j)$ we wish to recover the mean matrix $[F_{i,j}]$.

Each of these five settings has been studied in many additional contexts, including word embedding [SCH15], Genomic Analysis [ZLWY14, ZHPA13], and more [AGH$^+$14].

As observed above, in all these settings, the observations are generated by low-rank matrices. In the rest of the paper we show that relatively few observations suffice to learn the underlying matrices.

## 1.2  Unified formulation

We first unify the five settings, facilitating the interpretation of their matrix norm as the number of observations, and allowing us to subsequently show that all models have essentially the same answer.

Let $\mathbb{R}^{k \times m}$ be the collection of $k \times m$ real matrices, and let $\mathbb{R}_r^{k \times m}$ be its subset of matrices with rank at most $r$. Both our lower and upper bounds for recovering $k \times m$ matrices depend only on the larger of $k$ and $m$. Hence without loss of generality we assume square model matrices of size $k \times k$.

Note that in the first four models, $X_{i,j}$ is the number of times pair $(i, j)$ was observed. For example, for the Poisson model, it is the number of times customer $i$ bought product $j$. Hence $||X||_1 = \sum_{i,j} X_{i,j}$ is the total number of observations. In collaborative filtering, $X_{i,j} = \mathbb{1}_{i,j} \cdot Y_{i,j}$ does not carry the same interpretation, still $||X||_1 = \sum_{i,j} X_{i,j}$ will concentrate around $p \cdot k^2 \cdot F^{\mathrm{avg}}$, where $F^{\mathrm{avg}} = \frac{1}{k^2} \sum_{i,j} F_{i,j}$. And since $F_{i,j} \in [0, 1]$, therefore $F^{\mathrm{avg}}$ is typically a constant, hence $||X||_1$ will be proportional to the expected number of observations, $pk^2$.

We therefore scale the matrix $M$ so that $M_{i,j} = E(X_{i,j})$ reflects the expected number of observations of pair $(i, j)$. We let $M$ be $n \cdot [p_{i,j}]$ for distribution matrices, $[\lambda_{i,j}]$ for Poisson parameters, $[p_{i,j}]$ for Bernoulli parameters, $[t \cdot p_{i,j}]$ for Binomial parameters, and $[p \cdot F_{i,j}]$ for collaborative filtering. Consequently, for all models $M = \mathbb{E}[X]$, hence $||M||_1 = ||\mathbb{E}[X]||_1$ is the expected number of observations.

Let $M \in \mathbb{R}_r^{k \times k}$ be an unknown model matrix for one of the five settings, and let $X \sim M$ be a resulting observation matrix. We would like to recover $M$ from $X$. A model estimator is a mapping $M^{\mathrm{est}} : \mathbb{R}^{k \times k} \to \mathbb{R}^{k \times k}$ that associates with an observation $X$ an estimated model matrix $M^{\mathrm{est}} := M^{\mathrm{est}}(X)$.

Different communities have used different measures for how well $M^{\mathrm{est}}$ approximates $M$. For example community detection concerns the recovery of labels, a criterion quite specific to this particular application. Many other works considered recovery in squared-error, or Forbenius, norm, but as argued in Section 1.4, this measure is less meaningful for the applications we consider.

Perhaps the most apt estimation measure, and the one we adopt, is $L_1$ distance, the standard Machine-Learning accuracy measure. $L_1$ distance arises naturally in numerous applications and is the main criterion used for learning distributions. Since $M$ has non-unitary norm, we define the loss of the estimate $M^{\mathrm{est}}$ as the normalized $L_1$ distance between $M$ and $M^{\mathrm{est}}$,

$$L(M^{\mathrm{est}}) := L_M(M^{\mathrm{est}}) := ||M^{\mathrm{est}} - M||_1 / ||M||_1.$$

Note that for distribution matrices, this reduces to the standard $L_1$ norm. Also, similar to total variation distance, $L(M^{\mathrm{est}})$ upper bounds the absolute difference between the expected and predicted number of occurrences of pairs $(i, j)$ in any subset $S \subseteq [k] \times [k]$, normalized by the total observations.

While the multitude of applications has drawn considerable amount of work on these latent variable model, a majority of work assumes that the number of samples are plenty, way beyond the information theoretic limit, or assumes more stronger assumptions on the model matrix then just low rank, thus limiting its applicability. We show recovery is possible with number of samples only linear in $k$ and near linear in $r$ in all these models and with no other assumptions on low rank matrix $M$.

## 1.3  Overview of the background, main results and techniques

Recovery of distribution-matrices (the first model) in $L_1$ distance was first addressed in [HKKV18]. They considered matrices $M$ that can be factored as $UWU^\intercal$ where $W$ is $\widehat{r} \times \widehat{r}$ and positive semi-definite, and $U$ is non-negative. They derived a polynomial-time algorithm that requires $\mathcal{O}(w_M k \widehat{r}^2/\epsilon^5)$ samples, where $w_M \geq \widehat{r}^2$, hence at best guaranteeing sample complexity $\mathcal{O}(k \widehat{r}^4/\epsilon^5)$, and potentially much higher. Note however that their definition applies only to positive semi-definite, and not general matrices $M$, and even then, $\widehat{r}$ is at least the rank of $M$, and can be significantly higher.

We first lower bound the loss of any estimator. An array of $kr$ elements can be viewed as a special case of a $k \times k$ matrix of rank $r$. Simply place the array's $kr$ elements in the matrix's first $r$ rows, and set the remaining rows to zero. A well-known lower bound for learning discrete distributions, or arrays of Poisson parameters, in $L_1$ distance [KOPS15, HJW15] therefore implies:

Recall that $X \sim M$ means that observation matrix $X$ was generated by an underlying model matrix $M$, where $\mathbb{E}X = M$. Unless specified otherwise, the results apply to all five models described in Section 1.1.

**Theorem 1.** *For any $k$, $r$, $\epsilon < 1$, and $M \in \mathbb{R}_r^{k \times k}$, let $X \sim M$ via the Poisson parameters or distribution matrix model. Then for any, possibly random, estimator $M^{est}$,*

$$\sup_{M \in \mathbb{R}_r^{k \times k}: ||M||_1 \leq kr/\epsilon^2} \mathbb{E}_{X \sim M}[L(M^{est}(X))] = \Omega(\epsilon).$$

The bound implies that achieving expected normalized $L_1$ error $< \epsilon$ requires expected number of observations $||M||_1 = \Omega(kr/\epsilon^2)$. Equivalently, any estimator incurs an expected normalized $L_1$ error at least $\Omega(\min\{\sqrt{kr/||M||_1}, 1\})$.

Our main result is a polynomial-time algorithm *curated SVD* that returns an estimate $M^{\text{cur}} := M^{\text{cur}}(X)$ that essentially achieves the above lower bound for all five models and all matrices $M$.

**Theorem 2.** *Curated SVD runs in polynomial time, and for every $k$, $r$, $\epsilon > 0$, and $M \in \mathbb{R}_r^{k \times k}$ with $||M||_1 \geq \frac{kr}{\epsilon^2} \cdot \log^2 \frac{r}{\epsilon}$, if $X \sim M$, then with probability $\geq 1 - k^{-2}$,*

$$L(M^{cur}(X)) = \mathcal{O}(\epsilon).$$

A few observations are in order. While [HKKV18] provides weaker guarantees and only for a special subclass of matrices, Curated SVD achieves essentially the lower bound for all matrices. It also holds for all five models. It recovers $M$ with $\mathcal{O}(kr \log^2 r)$ observations. This number is linear in the large matrix dimension $k$, and near linear in the typically small rank $r$. This is the first such result for general matrices.

In many applications, only a small number of observations is available per row and column. For example on average each viewer may rate only few movies, and a person typically has few friends. Hence the number of observations is near linear in the dimension $k$. Our results are the first to enable learning general low rank matrices in these regimes.

With $n$ samples, general discrete distributions over $k$ elements can be learned to $L_1$ distance $\Theta(\sqrt{k/n})$. Theorems 1 and 2 show that essentially the same result holds for $L_1$ learning of low-rank matrices. The number of parameters is $kr$, the number of observations is $||M||_1$, and the normalized $L_1$ error is between $\sqrt{kr/||M||_1}$ and $\sqrt{kr/||M||_1} \log(r||M||_1/k)$.

To obtain these results we generalize a recent work [LLV17] that bounds the spectral distance between $M$ and $X$. This bound requires the strong condition that each entry of the corresponding Bernoulli parameter matrix $M$ should be within a constant factor from the average. The paper asked whether such results can be achieved for more general sparse graphs, possibly with the aid of regularization. We provide a counter example showing that such strong guarantees cannot hold for sparse graphs.

Instead, we derive a new spectral result (Theorem 3) that helps recover $M$ even when few observations are available, and may be of independent interest. The result applies to all models, not just Bernoulli, and shows that even when the number of observations is only linear in $k$, zeroing out a small submatrix in $M$ and $X$, and then regularizing the two matrices, results in a small spectral distance between them. This result is the first result to imply nontrivial concentration in spectral norm in linear sample regime for the general matrices in any of these settings. We believe this new result could potentially imply learning in the sparse sample regime for other settings that are not explicitly considered here.

**Theorem 3 (Informal, full version in Section 3)** There is a small unknown set of rows that when zeroed out from regularized versions of $X$ and $M$ results in small spectral distance between them.

Although this set of rows is unknown, we derive an algorithm that recovers $M$ to a small $L_1$ distance.

**Curated SVD (Informal, full version in Section 4)** Successively zeroes out few suspicious rows, ensuring that any small set of rows of $X$ don't have too large an influence on the recovered matrix.

We show that Theorem 3 implies that curated SVD achieves the recovery guaranties in Theorem 2.

## 1.4 Implications of the results and related work

As mentioned in the introduction, many communities have considered recovering model matrices from samples. Here we describe a few more of the most relevant work to this paper.

In community detection the goal is to infer communal structure from pairwise interactions between individuals. Much work has focused on the Stochastic Block models (SBM) where individuals fall in few communities $C_1, \ldots, C_r$ and the interaction probabilities $p_{i,j}$ are one constant if $i$ and $j$ are in the same community, and a different constant if $i$ and $j$ are in two distinct communities. Precise guarantees were provided for both exact recovery and detection [Abb17, AS15, ABH15, BLM15], even for the sparse regime. However, note that SBM only allows matrices of interaction probabilities that are a very special case of low rank matrices and not general low rank matrices.

A more general Mixed membership SBM associates each individual with an unknown $r$-dimensional vector reflecting weighted membership in each of $r$ communities, and the pairwise interaction probability is determined by inner product of the respective membership vectors. The resulting interaction probability matrix is a rank-$r$ Bernoulli parameter matrix. Recently, [HS17] considered a Bayesian setting where the resulting membership weights are both sparse and evenly distributed, and achieved weak detection with only $\mathcal{O}(kr^2)$ samples.

The additional assumptions needed in [HS17] imply that the entries in the resulting model matrix are within a constant factor from their average. By contrast our analysis applies to all low rank matrices $M$, and for $L_1$ recovery using $\mathcal{O}(kr \log^2 r)$ samples. We note that the goal in community detection setting is somewhat more specific than ours, and the two guarantees may not be directly comparable.

Another recent work [MD19] considered recovering Poisson and distribution matrices under the Frobenius norm. They showed that matrices $M$ with moderately-sized entries, can be recovered with $O(k \log^{3/2} k)$ samples, but the error depended on $M$'s maximum row and column sum. However, they note that the Frobenius norm "might not always be the most appropriate error metric" and point out that the $L_1$ norm is "much stronger" for these settings. A similar sentiment about $L_1$ is echoed by [HKKV18] in relation to the spectral norms. By contrast, our results apply to the $L_1$ norm, grow only linearly with $k$, and the error depends on $M$'s average, not the maximum, row and column sum.

Our collaborative filtering setting uses the same general bounded noise model as [BCLS17]. However they assume that the mean matrix $F$ is generated by a Lipschitz latent variable model. They recover $F$ to mean square error $\sum_{i,j}(F_{i,j} - \hat{F}_{i,j})^2/k^2 = \mathcal{O}(r^2/(pk)^{2/5})$, implying that recovering $F$ requires $pk^2 = \mathcal{O}(r^5k)$ samples. They also provides a nice survey of related matrix completion and show that their result is the first to achieve linear in $k$ recovery for the general bounded noise model.

By contrast, we make no additional assumptions, and recover $F$ to $L_1$ distance with $\mathcal{O}(kr \log^2 r)$ samples. Note also that $0 \leq F_{i,j} \leq 1$, hence $|F_{i,j} - \hat{F}_{i,j}| \geq |F_{i,j} - \hat{F}_{i,j}|^2$. Therefore normalized $L_1$ error upper bounds mean squared error. In Appendix D we show that our estimator achieves a better error bound, $\sum_{i,j}|F_{i,j} - \hat{F}_{i,j}|/k^2 = \mathcal{O}(\sqrt{r/pk} \cdot \log(rpk))$, even for the stronger $L_1$ norm.

Learning latent variables models has been addressed in several other communities that typically focused on computational efficiency when the data is in abundance, includes work on Topic Modelling [AGH+13, KW17, BBW18], and Hidden Markov Models [HKZ12, MR05, AGH+14], word embedding [SCH15], and Gaussian mixture models [Das99, GHK15, VW04].

## 1.5 Arrangement of the paper

The reminder of the paper is organised as follows. Section 2 defines some useful notations and recalls some useful properties for these models. Section 3 defines the regularization and establish bounds on

the regularized spectral distance between $X$ and $M$. Section 4 describes the Curated-SVD algorithm to recover $M$ and gives an overview of its analysis.

## 2  Preliminaries

### 2.1  Notation

We will use the following formulation for rank, singular values, and decompositions of a matrix. Every matrix $A \in \mathbb{R}^{k \times m}$ can be expressed in terms of its *singular-value decomposition (SVD)*, $A = \sum_{i=1}^{\min\{k,m\}} \sigma_i u_i v_i^{\mathsf{T}}$, where the *singular values* $\sigma_i := \sigma_i(A)$ are non-increasing and non-negative, and the right singular vectors $v_i := v_i(A)$ are orthogonal, as are the left singular vectors $u_i := u_i(A)$. A matrix has rank $r$ iff its first $r$ singular values are positive, and the rest are zero. A *t-truncated* SVD of a matrix is one where only the first $t$ singular values in the SVD are retained and the rest are discarded. For $t \leq \min\{k,m\}$, let $A^{(t)}$ denote the $t$-truncated SVD of a matrix $A \in \mathbb{R}^{k \times m}$, it is easy to see that the rank of $A^{(t)}$ is $\min(t, r)$.

The $L_1$ *"entry-wise" norm*, or $L_1$ *norm*, of a matrix $A$ is $||A||_1 := \sum_{i,j} |A_{ij}|$. Let $||A_{i,*}||_1 := \sum_j |A_{i,j}|$ and $||A_{*,j}||_1 := \sum_i |A_{i,j}|$ denote the $L_1$ norm of $i^{th}$ row and $j^{th}$ column of $A$, respectively.

The $L_2$ *norm* of a vector $v = (v(1), \ldots, v(m)) \in \mathbb{R}^m$ is $||v|| := \sqrt{\sum_{i=1}^m v(i)^2}$. The *spectral norm*, or *norm* for short, of a matrix $A \in \mathbb{R}^{k \times m}$ is $||A|| := \max_{v \in \mathbb{R}^m : ||v||=1} ||Av|| = \max_{u \in \mathbb{R}^k : ||u||=1} ||A^{\mathsf{T}} u||$.

### 2.2  A unified framework

We first describe a unified common framework for all five problems.

To unify distribution-matrices with Poisson-parameter matrices, we apply the well-known *Poisson trick* [Szp01], where instead of a fixed sample size $n$, we take $\mathrm{Poi}(n)$ samples. The resulting random variables $X_{i,j}$ will be Poisson and independent. Furthermore, since with probability $\geq 1 - 1/n^3$, the difference between $n$ and $\mathrm{Poi}(n)$ is $O(\sqrt{n \log n})$, this modification contributes only smaller order terms, and the algorithm and guarantees for Poisson parameters carry over to distribution matrices.

Having unified distribution- and Poisson-parameter- matrices, we focus on the remaining four settings. In all of them, the observations $X_{i,j}$ are independent non-negative random variables with $\mathbb{E}[X_{i,j}] = M_{i,j}$ and $\mathrm{Var}(X_{i,j}) \leq M_{i,j}$. The last inequality clearly holds for the Poisson, Bernoulli, and Binomial matrices. For collaborative filtering it follows as

$$\mathrm{Var}(X_{i,j}) \leq E((X_{i,j})^2) \leq E(X_{i,j}) = M_{i,j}.$$

Define the *noise matrix* $N := X - M$ as the difference between $X$ and its expectation $M$. Note that the spectral distance between $X$ and $M$ is the same as the the spectral norm of the noise matrix $N$.

Let $n_{\mathrm{avg}} := ||M||_1/k$ denote the average expected number of observations in each row and column. For simplicity, we assume $||M||_1$, the total number of expected observation, is known. Otherwise, since $\mathbb{E}[||X||_1] = ||M||_1$, it can be estimated very accurately.

## 3  Spectral norm of the regularized noise matrix

Recall that the noise matrix $N = X - M$, and its spectral norm $||N||$ is the spectral distance between observation matrix $X$ and $M$. When $||N|| = \mathcal{O}(\sqrt{n_{\mathrm{avg}}})$, a simple truncated-SVD of $X$ can be shown to recover $M$. Unfortunately $||N||$ can be $>> \sqrt{n_{\mathrm{avg}}}$, and one of the reason for this is when some row- or column-sums of $M$ far exceed the average. This is because the expected squared norm of row $i$ of $N$ is $\sum_j \mathrm{Var}(X_{i,j})$ which is roughly $||M_{i,*}||_1$, and if for some row $i$ of $M$, the row sum $||M_{i,*}||_1$ is much larger than the average value $n_{\mathrm{avg}}$ across the rows then $||N||$ will be large as well. Similarly for the columns.

The difficulty caused by the heavy rows and columns of $M$ can be mitigated by a regularization that reduces their weight. Let $w = (w^f, w^b)$ where $w^f = ((w^f(1), .., w^f(k))$ and $w^b = (w^b(1), .., w^b(k))$

are the row and columns regularization weights, all at least 1. And let $D(u)$ be the diagonal matrix with entries $u(i)$. The *w-regularized* $A \in \mathbb{R}^{k \times k}$ is

$$R(A, w) := D^{-\frac{1}{2}}(w^f) \cdot A \cdot D^{-\frac{1}{2}}(w^b).$$

Upon multiplying the $i^{th}$ row of matrix $A$ by $(w^f(i))^{-1/2}$, its expected squared norm reduces by a factor $1/w^f(i)$, and similarly for the columns. Therefore, selecting regularization weights $\tilde{w} = (\tilde{w}^f, \tilde{w}^b)$, where $\tilde{w}^f(i) = \max\{1, ||M_{i,*}||_1/n_{\text{avg}}\}$ and $\tilde{w}^b(j) = \max\{1, ||M_{*,j}||_1/n_{\text{avg}}\}$ would reduce the expected squared norm of heavy rows and columns of $R(N, \tilde{w})$ to $n_{\text{avg}}$.

Unfortunately, the $||M_{i,*}||_1$ and $||M_{*,j}||_1$'s are not known, hence neither is $\tilde{w}$. However, $\mathbb{E}[\sum_j X_{i,j}] = ||M_{i,*}||_1$, hence we approximate $\tilde{w}$ by $\bar{w} = (\bar{w}^f, \bar{w}^b)$, where

$$\bar{w}^f(i) := \max\{1, \tfrac{||X_{i,*}||_1}{n_{\text{avg}}}\} \quad \text{and} \quad \bar{w}^b(j) := \max\{1, \tfrac{||X_{*,j}||_1}{n_{\text{avg}}}\}.$$

Unless specified otherwise we use weights $\bar{w}$, and refer to them as *weights*. This is one of the several commonly used regularizations in spectral methods for community detection [CCT12, QR13, JY13].

When $n_{\text{avg}} = o(\log k)$, some rows and columns may have regularization weights that are below the ideal weights, $\bar{w}^f(i) \ll \tilde{w}^f(i)$ and $\bar{w}^b(j) \ll \tilde{w}^b(j)$, and will not be regularized properly. Yet as shown in Theorem 3, our technique can handle these problematic rows and columns as well.

The *regularized spectral distance* between two matrices $A$ and $B$ is $||R(A - B, w)||$, the spectral norm of their regularized spectral difference. Lemma 4 in the next section relates $||R(N, \bar{w})||$ to $L_1$ recovery guarantees, and implies that when $||R(N, \bar{w})|| \leq \mathcal{O}(\sqrt{n_{\text{avg}}})$ a simple variation of truncated SVD can recover $M$ from $X$ to the minmax lower bound on $L_1$ recovery in Theorem 1.

When the number of samples is at least a few $\log k$ factors more than $k$, this spectral concentration could probably be achieved with the help of the regularization. The more interesting, challenging, and prevalent setting, is when few observations are available, and this is the main focus of the paper.

For Bernoulli Parameter matrices, [LLV17] obtained the tight bound $||R(N, \bar{w})|| = \mathcal{O}(\sqrt{n_{\text{avg}}})$ that holds even for sparse graphs, or equivalently few observations. However it requires that every $M_{i,j} = \mathcal{O}(||M||_1/k^2)$, hence holds only for a very limited and often impractical subclasses of parameter matrices $M$. They posed the question whether this bound also holds for general $M$. In Appendix F, we provide a counterexample that answers the question in the negative. We construct an explicit Bernoulli parameter matrix $M$, s.t. w.h.p. $||R(N, \bar{w})|| = \Omega(n_{\text{avg}})$, much larger than $\mathcal{O}(\sqrt{n_{\text{avg}}})$.

Yet upper bounding $||R(N, \bar{w})||$ is just one approach to achieving optimal sample complexity. One of this paper's contribution is an alternative approach that decomposes the noise matrix $N$ into two parts. A large part with small spectral norm, and a small part, in fact a submatrix, that may have a large spectral norm. While we cannot identify the "noisy" part, as shown in Section 4, we can ensure that no small part has a large influence on the estimate.

The next theorem establishes the above partition for all parameter matrices, and for all settings. To specify the matrix decomposition, both here and later, for $A \in \mathbb{R}^{k \times m}$ and subset $S \subseteq [k] \times [m]$, let $A_S$ be the projection of matrix $A$ over $S$ that agrees with $A$ for indices in $S$ and is zero elsewhere. Further let $A_I := A_{I \times [m]}$ and $A_{I^c} := A_{[k] \setminus I \times [m]}$ be the matrices derived from $A$ by zeroing out all rows outside, and inside, set $I$, respectively.

For the weights $\bar{w} = (\bar{w}^f, \bar{w}^b)$ above, let $\bar{w}^f(I) := \sum_{i \in I} \bar{w}^f(i)$ denote the *weight* of row subset $I$. In the next theorem and thereafter, we refer to rows that are needed to be zero out to achieve spectral concentration as *contaminated rows*.

**Theorem 3.** *For $X \sim M$, any $\epsilon \geq \frac{1}{n_{\text{avg}}} \max\left(\frac{\log^4 k}{k}, \exp^{-\frac{n_{\text{avg}}}{8}}\right)$, with probability $\geq 1 - 6k^{-3}$, there is a row subset $I_{\text{cn}} \subseteq [k]$ of possibly contaminated rows with weight $\bar{w}^f(I_{\text{cn}}) \leq \epsilon k$ and*

$$||R(N, \bar{w})_{I_{\text{cn}}^c}|| \leq \mathcal{O}\left(\sqrt{n_{avg}} \cdot \log \tfrac{2}{\epsilon}\right).$$

Since $\bar{w}^f(i)$ is the maximum of 1 and the number of observations in row $i$, the theorem shows that for some $I_{\text{cn}} \subseteq [k]$ with only a few rows, $X_{I_{\text{cn}}}$ contains only few observations, and zeroing out rows $I_{\text{cn}}$ from the regularized noise matrix $R(N, \bar{w})$ would result in small spectral norm. The above result is the first to imply nontrivial concentration in spectral norm, in linear sample regime, for the general

matrices in any of the settings we considered. We derive an algorithm that uses this result and to provably recovers low rank parameter matrices, even when the number of samples are only $\mathcal{O}(k)$.

To prove Theorem 3, we extend a technique used in [LLV17] for specialized Bernoulli matrices with entries all below $\mathcal{O}(||M||_1/k^2)$, to bound the spectral noise norm of general models and Matrices. In Appendix E, we use standard probabilistic methods and concentration inequalities, to establish concentration in $\ell_\infty \to \ell_2$ norm for all sub-matrices of $R(N, \bar{w})$. We then recursively apply a form of Grothendieck-Pietsch Factorization [LT13] and incorporate these bounds to partition $R(N, \bar{w})$ into successively smaller submatrices and upper bound their spectral norms, until the resulting submatrix is very small. Finally we show that the squared spectral norm of any matrix is at most the sum of the squared spectral norms of its decomposition parts, and thus upper bound the spectral norm of $R(N, \bar{w})$, except for the small submatrix, that is excluded.

## 4    Recovery Algorithms

We start by describing a simple generalisation of truncated SVD, on which our algorithm builds. Recall that for any $A \in \mathbb{R}^{k \times k}$, $R(A, w)$ denotes the regularized matrix $A$, and that $R(A, w)^{(t)}$ is its rank-$t$ truncated SVD. For any $t > 0$, regularization weights $w$, and matrix $A$, let the $(t, w)$-SVD of $A$ be the de-regularized, rank-$t$-truncated SVD of regularized matrix $A$,

$$A^{(t,w)} := D^{\frac{1}{2}}(w^f) \cdot R(A, w)^{(t)} \cdot D^{\frac{1}{2}}(w^b).$$

Let $A$ be rank-$r$ and $B$ be any matrix. The next lemma bounds the $L_1$ distance between $A$ and $B^{(r,w)}$ in terms of the regularized spectral distance between $A$ and $B$. Appendix G provides a simple proof.

**Lemma 4.** *For any rank-$r$ matrix $A \in \mathbb{R}_r^{k \times k}$, matrix $B \in \mathbb{R}^{k \times k}$, and weights $w = (w^f, w^b)$,*

$$||A - B^{(r,w)}||_1 \leq \sqrt{r \cdot (\sum_i w^f(i))(\sum_j w^b(j))} \cdot ||R(A - B, w)||.$$

Recall that $N = X - M$. The lemma implies that when $||R(N, \bar{w})||$ is small, $X^{(r,\bar{w})}$, the $(r, \bar{w})$-SVD of $X$, would recover $M$. However, $||R(N, \bar{w})||$ could be large. Instead, Theorem 3 in the last section implies that w.h.p., the following *essential property* holds.

**Essential property**    There is an unknown *contaminated* set, $I_{\text{cn}} \subset [k]$, such that after zeroing all $I_{\text{cn}}$ rows in $X$ and $M$, their regularized spectral distance is at most,

$$||R(N, \bar{w})_{I_{\text{cn}}^c}|| \leq \tau := \mathcal{O}(\sqrt{n_{\text{avg}}} \log(r n_{\text{avg}})), \tag{1}$$

and the weight of the set $I_{\text{cn}}$ is at most, $\bar{w}^f(I_{\text{cn}}) \leq W_{\text{cn}} := \mathcal{O}(k/(r n_{\text{avg}})^2)$. The above property holds with probability $\geq 1 - 6k^{-3}$, by choosing $\epsilon = 1/(r n_{\text{avg}})^2$ in Theorem 3. As the property holds with high probability, the reminder of this section assumes that it holds.

This property implies a small regularized spectral distance between $X_{I_{\text{cn}}^c}$ and $M_{I_{\text{cn}}^c}$, and since $\bar{w}^f(I_{\text{cn}}) \leq W_{\text{cn}}$, the "noisier" part of observation matrix, namely $X_{I_{\text{cn}}}$, is limited to just a few rows and observations.

Recall that the $(r, w)$-SVD of any matrix $A$ is the de-regularized rank-$r$-truncated SVD of regularized matrix $R(A, w)$. Equation (1) and Lemma 4 implies that a simple $(r, \bar{w})$-SVD of $X_{I_{\text{cn}}^c}$ would recover $M_{I_{\text{cn}}^c}$ to a small error, and since the rows $I_{\text{cn}}$ of $X$ have only a few observations, recover $M$ as well. But the set $I_{\text{cn}}$ of contaminated rows is unknown.

Building on this simple $(r, \bar{w})$-SVD, we derive our main algorithm, Curated SVD, that achieves the same performance guarantee up to a constant factor, even when the set $I_{\text{cn}}$ is unknown.

In Curated SVD, for every row subset $I \subset [k]$ with weight $\bar{w}^f(I) \leq W_{\text{cn}}$, we limit the maximum impact the submatrix $R(X, \bar{w})_I$ can have on the truncated SVD of regularized observation matrix $R(X, \bar{w})$. In particular, this limits the impact of the unknown "noisier" submatrix $R(X, \bar{w})_{I_{\text{cn}}}$.

To describe the algorithm, first we formally define the *impact* of a row subset on SVD components. For a general matrix $A \in \mathbb{R}^{k \times k}$, let $A = \sum_{j=1}^k \sigma_j u_j v_j^\intercal$ be its SVD. Then for any $i, j \in [k]$, let the *impact* of row $i$ of $A$, on the $j^{th}$ component in SVD of $A$, be $\mathcal{H}(A, i, j) := \sigma_j^2 u_j(i)^2$. Similarly,

for a row subset $I$, the *impact* of $A_I$, or simply impact of $I$, on $j^{th}$ component in SVD of $A$ be $\mathcal{H}(A, I, j) := \sum_{i \in I} \mathcal{H}(A, i, j)$, the sum of the impact of each row. From the standard properties of SVD, it is easy to see that $\mathcal{H}(A, I, j) = \sigma_j^2 \sum_{i \in I} u_j(i)^2 = ||A_I \cdot v_j||^2$.

Next, we present an overview of the algorithm and its analysis. Essentially, Curated SVD finds a set $I_{\mathrm{zr}} \subseteq [k]$ of row indices, such that the following *objectives* are fulfilled:

**(i)** Let $(R(X, \bar{w})_{I_{\mathrm{zr}}^{\mathrm{c}}})^{(2r)} = \sum_{j=1}^{2r} \sigma_j u_j v_j^{\mathsf{T}}$, be the rank $2r$-truncated SVD of regularized observation matrix upon zeroing out rows $I_{\mathrm{zr}}$. For every row subset $I$ of weight $\bar{w}^f(I) \leq W_{\mathrm{cn}}$ and $\forall\, j \in [2r]$, the impact of $I$ on the $j^{th}$ component is $\mathcal{H}(R(X, \bar{w})_{I_{\mathrm{zr}}^{\mathrm{c}}}, I, j) = \sigma_j^2 \sum_{i \in I} u_j(i)^2 \leq 16\tau^2$.

**(ii)** The total weight of the set $I_{\mathrm{zr}}$ is small, $\bar{w}^f(I_{\mathrm{zr}}) = \sum_{i \in I_{\mathrm{zr}}} \bar{w}^f(i) \leq \mathcal{O}(k/n_{\mathrm{avg}})$.

After finding such a row subset $I_{\mathrm{zr}}$, Curated SVD zeroes out rows $I_{\mathrm{zr}}$ from $X$ and simply returns the $(2r, \bar{w})$-SVD of $X_{I_{\mathrm{zr}}^{\mathrm{c}}}$ as the estimate of $M$.

Objective **(i)**, ensures that for every collection of rows $I$ of weight $\bar{w}^f(I) \leq W_{\mathrm{cn}}$, the impact of $R(X, \bar{w})_I$ on each of the first $2r$ components of SVD is small. As weight of $I_{\mathrm{cn}}$ is at most $W_{\mathrm{cn}}$, in particular it limits the impact of the noisier submatrix $R(X, \bar{w})_{I_{\mathrm{cn}}}$.

Objective **(ii)** ensures that the weight of $I_{\mathrm{zr}}$, and hence the number of observations in $X_{I_{\mathrm{zr}}}$, that get ignored in final truncated regularized SVD is small. This limits the loss due to the ignored rows $I_{\mathrm{zr}}$.

Lemma 13 in the appendix uses these two observations to show that, the $(2r, \bar{w})$-SVD of $X_{I_{\mathrm{zr}}^{\mathrm{c}}}$ recovers $M$.

Next, we describe the algorithm Curated-SVD and show that it finds a set $I_{\mathrm{zr}}$ that achieves Objective **(i)** and **(ii)**. The pseudo-code of the algorithm is in Appendix B.

## 4.1   Description of the Curated SVD

Curated-SVD starts with $I_{\mathrm{zr}} = \phi$. In each iteration it calculates $\left(R(X, \bar{w})_{I_{\mathrm{zr}}^{\mathrm{c}}}\right)^{(2r)} = \sum_{j=1}^{2r} \sigma_j u_j v_j^{\mathsf{T}}$, the rank-$2r$ truncated SVD of $R(X, \bar{w})_{I_{\mathrm{zr}}^{\mathrm{c}}}$. Then it calls subroutine *Row-Deletion* for each $j \in [2r]$.

Row deletion checks for the row subsets $I \subseteq I_{\mathrm{zr}}^{\mathrm{c}}$ with small weight and high impact that violate Objective **(i)**. It then adds rows from such subsets $I$ to $I_{\mathrm{zr}}$ to reach Objective **(i)** in a way that $I_{\mathrm{zr}}$ does not end up too heavy. To do this, Row-Deletion tries to find a row subset $I \subseteq I_{\mathrm{zr}}^{\mathrm{c}}$ with weight $\bar{w}^f(I) \leq W_{\mathrm{cn}}$ and maximum impact $\mathcal{H}(R(X, \bar{w})_{I_{\mathrm{zr}}^{\mathrm{c}}}, I, j) = \sigma_j^2 \sum_{i \in I} u_j(i)^2$. This however is essentially the well-known NP-hard 0-1-knapsack problem. Instead, we use a greedy 0.5-approximation algorithm [SCGDS92] for 0-1 knapsack, to obtain a row subset $I$ with weight $\leq W_{\mathrm{cn}}$, such that its impact is at least half of the maximum possible, namely

$$\mathcal{H}(R(X, \bar{w})_{I_{\mathrm{zr}}^{\mathrm{c}}}, I, j) \geq 0.5 \max\{\mathcal{H}(R(X, \bar{w})_{I_{\mathrm{zr}}^{\mathrm{c}}}, I', j) : I' \subseteq I_{\mathrm{zr}}^{\mathrm{c}}, \textstyle\sum_{i \in I'} \bar{w}^f(i) \leq W_{\mathrm{cn}}\}.$$

If the impact of row collection $I$, found using the greedy algorithm, is $\mathcal{H}(R(X, \bar{w})_{I_{\mathrm{zr}}^{\mathrm{c}}}, I, j) \leq 8\tau^2$, sub-procedure Row-Deletion terminates. Else it adds a row $i \in I$ to $I_{\mathrm{zr}}$, with probability of row $i \in I$ proportional to its impact to the weight ratio, $\mathcal{H}(R(X, \bar{w})_{I_{\mathrm{zr}}^{\mathrm{c}}}, i, j)/\bar{w}^f(i)$. Row-Deletion repeats this procedure on the remaining rows in $I_{\mathrm{zr}}^{\mathrm{c}}$, until it terminates.

After calling Row-Deletion for each $j \in [2r]$, Curated-SVD checks if the new rows were added to $I_{\mathrm{zr}}$ in this iteration, in which case it repeats the same procedure in the next iteration with updated $I_{\mathrm{zr}}$, else it returns $(2r, \bar{w})$-SVD of $X_{I_{\mathrm{zr}}^{\mathrm{c}}}$ as the estimate of $M$.

**Curated SVD achieves Objective (i).**   Iterations in Curated-SVD stop when for the current choice of $I_{\mathrm{zr}}$, for all $j \in [2r]$, Row-Deletion fails to add any row to $I_{\mathrm{zr}}$. This happens when for each $j \in [2r]$, the greedy-approximation algorithm in Row-Deletion finds the row subset $I$ that has impact $\leq 8\tau^2$, which implies that the impact of every row subsets $I' \subseteq I_{\mathrm{zr}}^{\mathrm{c}}$ of weight $\bar{w}^f(I') \leq W_{\mathrm{cn}}$, is at most $16\tau^2$. Therefore, iterations in the algorithm stops only when $I_{\mathrm{zr}}$ meets Objective **(i)**.

**Curated SVD achieves Objective (ii) w.p. $1 - \mathcal{O}(k^{-2})$.**   This is more challenging of the two objectives. The proof is based on the following key observation proved in the Appendix C.1.

There exist a row subset $I_{\mathrm{hv}}$ s.t. for every subset $I \subseteq I_{\mathrm{hv}}^{\mathrm{c}}$ of weight $\bar{w}^f(I) \leq W_{\mathrm{cn}}$, the following holds $||R(X_I, \bar{w})|| \leq 2\tau$. And moreover, the weight of $I_{\mathrm{hv}}$ is $\bar{w}^f(I_{\mathrm{hv}}) \leq \mathcal{O}(k/n_{\mathrm{avg}})$.

It is easy to see that $||R(X_I, \bar{w})||^2$ upper bounds the impact submatrix $R(X_I, \bar{w})$ can have on the components of SVD. Therefore, from the previous observation, any row subset $I$ of weight $\bar{w}^f(I) \leq W_{cn}$ with impact $\geq 8\tau^2$, must have more than half of its impact due to the rows $I \cap I_{hv}$. Recall that Row-Deletion adds a row from $I$ to $I_{zr}$ only when $I$ has impact $\geq 8\tau^2$. We show that in each step in expectation it adds more weight from $I \cap I_{hv}$ to $I_{zr}$, then from $I \cap I_{hv}^c$.

Using this we show that, the expected total weight of rows added to $I_{zr}$ is at most a constant times the weight of $I_{hv}$, that is $\mathcal{O}(k/n_{avg})$, and Objective (ii) is achieved with constant probability. Repeating the procedure $\mathcal{O}(\log k)$ times on the same data, Objective (ii) holds with probability $1 - \mathcal{O}(k^{-2})$.

Finally, by combining Lemma 13 and the fact that the Curated SVD achieves both objectives, we prove Theorem 2 in Appendix D.

## Broader impact

This work does not present any foreseeable societal consequence.

## Acknowledgements

We thank Vaishakh Ravindrakumar and Yi Hao for their helpful comments in the prepration of this manuscript.

We are grateful to the National Science Foundation (NSF) for supporting this work through grants CIF-1564355 and CIF-1619448.

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
