[Supplementary Material]

# A  Preliminaries

This section contains some standard linear-algebra results used in the proofs. It shares terminology with Section 2. The *Frobenius*, or *entry-wise $L_2$, norm* of a matrix $A$ is

$$||A||_F := \sqrt{\sum_{i,j}|A_{ij}|^2}.$$

**Theorem 5.** (Singular value decomposition [Bha13]) *For all $A \in \mathbb{R}^{k \times m}$,*

1. *$Av_i(A) = \sigma_i(A)u_i(A)$ and $A^\intercal u_i(A) = \sigma_i(A)v_i(A)$, $\forall i$.*

2. *$||A|| = \sigma_1(A)$.*

3. *(Courant-Fischer Theorem for Singular Values):*

$$\sigma_i(A) = \min_{S:dim(S)=m-i+1} \max_{v \in S:||v||=1} ||Av|| = \max_{S:dim(S)=i} \min_{v \in S:||v||=1} ||Av||.$$

The singular values of a submatrix are smaller than those of the original matrix. In particular, the same holds for their spectral norms.

**Theorem 6.** (Interlacing property of Singular Values [Que87]) *For any $A \in \mathbb{R}^{k \times m}$, $I \subseteq [k]$, $J \subseteq [m]$, and $i \leq \min(k,m)$,*

$$\sigma_i(A_{I \times J}) \leq \sigma_i(A).$$

Singular values are subadditive.

**Theorem 7.** *(Weyl's Inequality for Singular Values [Bha13]) For all $A, A' \in \mathbb{R}^{k \times m}$ and $i, j \geq 1$ s.t. $i + j - 1 \leq \min(k,m)$,*

$$\sigma_{i+j-1}(A + A') \leq \sigma_i(A) + \sigma_j(A').$$

# B  Pseudo Code for the Algorithm

---
**Algorithm 1** CURATED-SVD

---
**Input** : Matrix $X$, $r$, $W_{\text{cn}}$ and $\tau$
**Output** : Matrix $\widehat{M}$
1:  $I^{\text{zr}} \leftarrow \phi$
2:  **while** true **do**
3:     $\mathcal{U} \leftarrow R(X, \bar{w})_{I_{\text{zr}}^{\text{c}}}$ {Recall $R(X, \bar{w}) = D^{-\frac{1}{2}}(\bar{w}^f) \cdot X \cdot D^{-\frac{1}{2}}(\bar{w}^b)$}
4:     Perform rank $2r$ truncated SVD on $\mathcal{U}$ to get $\mathcal{U}^{(2r)} = \sum_{j=1}^{2r} \sigma_j u_j v_j^\intercal$
5:     $I_{\text{zr}}^{\text{old}} \leftarrow I_{\text{zr}}$
6:     **for** $j \in [2r]$ **do**
7:        ROW DELETION($\sigma_j \cdot u_j$, $\bar{w}^f$, $I_{\text{zr}}$, $8\tau^2$, $W_{\text{cn}}$) {to update $I_{\text{zr}}$}
8:     **end for**
9:     **if** $I_{\text{zr}}^{\text{old}} == I_{\text{zr}}$ **then**
10:       Break;
11:    **end if**
12: **end while**
13: $M^{\text{cur}} \leftarrow D^{\frac{1}{2}}(\bar{w}^f) \cdot \mathcal{U}^{(2r)} \cdot D^{\frac{1}{2}}(\bar{w}^f)$ {Same as $(2r, \bar{w})$-SVD of $X_{I_{\text{zr}}^{\text{c}}}$}
14: Return( $M^{\text{cur}}$ )

---

**Algorithm 2** ROW DELETION

**Input** : $u = \sigma_j \cdot u_j$, $w = \bar{w}^f$, $I_{\mathrm{zr}}$, $V = 8\tau^2$, $W = W_{\mathrm{cn}}$
**Output**: Updated $I_{\mathrm{zr}}$

1: **while** True **do**
2:     Use 0.5-approx Algorithm [SCGDS92] for 0-1 knapsack problem to find subset $I \subseteq I_{\mathrm{zr}}^{\mathrm{c}}$ with total weight $\sum_{i \in I} w(i) \leq W$ and
    impact $\sum_{i \in I} u(i)^2 \geq 0.5 \max\{\sum_{i \in I'} u(i)^2 : I' \subseteq I_{\mathrm{zr}}^{\mathrm{c}}, \sum_{i \in I'} w(i) \leq W\}$
    {*Comment: Recall that impact* $\mathcal{H}(R(X, \bar{w})_{I_{\mathrm{zr}}^{\mathrm{c}}}, I, j) = \sum_{i \in I} \sigma_j^2 u_j^2(i) = \sum_{i \in I} u(i)^2$ }
3:     **if** $\sum_{i \in I} u(i)^2 \leq V$ **then**
4:         Return;
        {*Comment: "if" condition ensures* $\max_{I' \subseteq I_{\mathrm{zr}}^{\mathrm{c}} : \sum_{i \in I'} w(i) \leq W_{\mathrm{cn}}} \mathcal{H}(R(X, \bar{w})_{I_{\mathrm{zr}}^{\mathrm{c}}}, I', j) \leq 2V$ }
5:     **end if**
6:     $I_{\mathrm{zr}} \leftarrow I_{\mathrm{zr}} \bigcup$ An element $i \in I$ where the probability of picking $i$ is proportional to $\frac{u(i)^2}{w(i)}$
7: **end while**

## C   Analysis of Curated SVD algorithm

Subsection C.1 shows that w.h.p. Curated SVD achieves Objective (ii). Subsection C.2 shows that if $I_{\mathrm{zr}}$ achieves both objectives then regularized Curated SVD recovers $M_{I_{\mathrm{cn}}^{\mathrm{c}}}$.

### C.1   Curated SVD achieves Objective (ii)

We first show that both the objectives can be achieved simultaneously.

**Lemma 8.** *Assume that essential property* (1) *holds. There is a row-collection* $I_{\mathrm{hv}}$ *of weight* $\sum_{i \in I_{\mathrm{hv}}} \bar{w} \leq \mathcal{O}(k/n_{avg})$, *such that for any subset* $I \subseteq I_{\mathrm{hv}}^{\mathrm{c}}$ *of weight* $\sum_{i \in I} \bar{w}^f(i) \leq W_{\mathrm{cn}}$,

$$\|R(X_I, \bar{w})\| \leq 2\tau.$$

**Implication of the above Lemma**   It is easy to see that for any subset $I \subseteq [k]$, $\|R(X_I, \bar{w})\|^2$ upper bounds the impact $R(X_I, \bar{w})$ can have on the SVD. Hence, if we let $I_{\mathrm{zr}} = I_{\mathrm{hv}}$ then both the objectives will be satisfied. However, the above lemma only shows the existence of $I_{\mathrm{hv}}$ with small weight, and it does not give a computationally efficient way to find $I_{\mathrm{hv}}$. We later show that w.h.p. Curated-SVD finds $I_{\mathrm{zr}}$, efficiently, with weight at most a constant times that of $I_{\mathrm{hv}}$.

In proving the above lemma, we use the following two auxiliary lemmas, which are stated for general matrices. The proofs of these two lemmas appear in the Appendix G, along with the proof of other Linear algebra results used in the paper.

**Lemma 9.** *For any matrix $A$ and weight vectors $w^f$ and $w^b$ with positive entries*

$$\|D^{-\frac{1}{2}}(w^f) \cdot A \cdot D^{-\frac{1}{2}}(w^b)\| \leq \sqrt{\max_i \frac{\|A_{i,*}\|_1}{w^f(i)} \times \max_j \frac{\|A_{*,j}\|_1}{w^b(j)}}.$$

Applying the lemma for $A = X$ and weights $w^f = \bar{w}^f$ and $w^b = \bar{w}^b$ we get:

$$\|R(X, \bar{w})\| \leq \sqrt{\max_i \frac{\|X_{i,*}\|_1}{\bar{w}^f(i)} \times \max_j \frac{\|X_{*,j}\|_1}{\bar{w}^b(j)}} \leq n_{\mathrm{avg}}. \tag{2}$$

Next, we state the second auxiliary lemma required in proving Lemma 8

**Lemma 10.** *Let $A$ be a $k \times m$ matrix such that $\sigma_1(A) \leq \alpha$ and $\sigma_{r+1}(A) \leq \beta$. Then the number of disjoint row subsets $I \subset [k]$ such that $\|A_I\| > 2\beta$ is at most $\left(\frac{r\alpha}{\beta}\right)^2$.*

Using the above two auxiliary lemmas, we prove Lemma 8.

*Proof of Lemma 8.* Using Weyl's inequality 7

$$\sigma_{r+1}(R(X, \bar{w})_{I_{\mathrm{cn}}^{\mathrm{c}}}) \leq \sigma_{r+1}(R(M, \bar{w})_{I_{\mathrm{cn}}^{\mathrm{c}}}) + \sigma_1(R(X - M, \bar{w})_{I_{\mathrm{cn}}^{\mathrm{c}}})$$

$$\overset{(a)}{\leq} \sigma_1(R(N,\bar{w})_{I_{\mathrm{cn}}^{\mathrm{c}}}) = ||R(N,\bar{w})_{I_{\mathrm{cn}}^{\mathrm{c}}}|| \leq \tau, \tag{3}$$

here (a) uses $X - M = N$, and the fact that $M$ has rank $r$.

From equation (2) and Theorem 6,

$$\sigma_1(R(X,\bar{w})_{I_{\mathrm{cn}}^{\mathrm{c}}}) \leq \sigma_1(R(X,\bar{w})) \leq n_{\mathrm{avg}}. \tag{4}$$

Applying Lemma 10 for $A = R(X,\bar{w})_{I_{\mathrm{cn}}^{\mathrm{c}}}$, and using (3) and (4) shows that the number of disjoint subsets $I \subset I_{\mathrm{cn}}^{\mathrm{c}}$ such that $||R(X_I,\bar{w})|| > 2\tau$ is at most $\left(\frac{rn_{\mathrm{avg}}}{\tau}\right)^2$.

Therefore, the number of disjoint subsets $I \subset I_{\mathrm{cn}}^{\mathrm{c}}$, such that $||R(X_I,\bar{w})|| > 2\tau$ and weight $\bar{w}^f(I) \leq W_{\mathrm{cn}}$, is also at most $\left(\frac{rn_{\mathrm{avg}}}{\tau}\right)^2$. Since every such subset $I$ has weight $\bar{w}^f(I) \leq W_{\mathrm{cn}}$, the combined total weight of all such disjoint subsets is $\leq W_{\mathrm{cn}} \cdot \left(\frac{rn_{\mathrm{avg}}}{\tau}\right)^2$.

Let $I_{\mathrm{hv}} \subseteq [k]$ be the subset formed by combining all these disjoint subsets, and $I_{\mathrm{cn}}$. It is clear from the construction of $I_{\mathrm{hv}}$ that, for every subset $I \subset I_{\mathrm{hv}}^{\mathrm{c}}$ with weight $\bar{w}^f(I) \leq W_{\mathrm{cn}}$,

$$||R(X_I,\bar{w})|| \leq 2\tau.$$

Finally, noting that the total weight of subset $I_{\mathrm{hv}}$ is at most $W_{\mathrm{cn}} \cdot \left(\frac{rn_{\mathrm{avg}}}{\tau}\right)^2 + \sum_{i \in I_{\mathrm{cn}}} \bar{w}^f(i) \leq W_{\mathrm{cn}} \cdot \left(\frac{rn_{\mathrm{avg}}}{\tau}\right)^2 + W_{\mathrm{cn}}$, and using $W_{\mathrm{cn}} = \mathcal{O}(k/(rn_{\mathrm{avg}})^2)$ and $\tau = \mathcal{O}(\sqrt{n_{\mathrm{avg}}}\log(rn_{\mathrm{avg}}))$ completes the proof. ∎

The next lemma bounds the expected weight of $I_{\mathrm{zr}}$, the set of rows zeroed out by Curated SVD, by showing that it is at most twice the weight of $I_{\mathrm{hv}}$.

**Lemma 11.** *Assume that essential property* (1) *holds. When Curated SVD terminates, the total weight of the final set of rows $I_{\mathrm{zr}}$ satisfy,*

$$\mathbb{E}\left[\bar{w}^f(I_{\mathrm{zr}})\right] \leq \mathcal{O}(k/n_{\mathrm{avg}}).$$

*Proof.* Recall that the rows get added to $I_{\mathrm{zr}}$ one by one. Let $Z_t$ denotes the weight of the $t^{th}$ row that gets added to $I_{\mathrm{zr}}$. Let indicator random variable $\mathbb{1}_t(I_{\mathrm{hv}}) = 1$, if the $t^{th}$ that gets added in $I_{\mathrm{zr}}$ belongs to $I_{\mathrm{hv}}$ and similarly indicator random variable $\mathbb{1}_t(I_{\mathrm{hv}}^{\mathrm{c}}) = 1$, if the $t^{th}$ that gets added in $I_{\mathrm{zr}}$ doesn't belongs to $I_{\mathrm{hv}}$. Clearly $Z_t = Z_t \cdot \mathbb{1}_t(I_{\mathrm{hv}}) + Z_t \cdot \mathbb{1}_t(I_{\mathrm{hv}}^{\mathrm{c}})$. We first show that for any $t$,

$$\mathbb{E}[Z_t \cdot \mathbb{1}_t(I_{\mathrm{hv}}^{\mathrm{c}})] \leq \mathbb{E}[Z_t \cdot \mathbb{1}_t(I_{\mathrm{hv}})].$$

Let $I_t \subseteq I_{\mathrm{zr}}^{\mathrm{c}}$ denote the row subset from which $t^{th}$ row added to $I_{\mathrm{zr}}$ was chosen, probabilistically, by Row-Deletion (Line 6 in Row-Deletion procedure). Since Row-Deletion chooses rows only from the row subsets that have weight $\leq W_{\mathrm{cn}}$ and impact $> 8\tau^2$ on one of the components of SVD, therefore, $\bar{w}^f(I_t) \leq W_{\mathrm{cn}}$ and some $j \in [2r]$, and its impact

$$\mathcal{H}(R(X,\bar{w})_{I_{\mathrm{zr}}^{\mathrm{c}}}, I_t, j) > 8\tau^2.$$

We decompose $\mathcal{H}(R(X,\bar{w})_{I_{\mathrm{zr}}^{\mathrm{c}}}, I_t, j)$ in two parts as

$$\mathcal{H}(R(X,\bar{w})_{I_{\mathrm{zr}}^{\mathrm{c}}}, I_t, j) = \mathcal{H}(R(X,\bar{w})_{I_{\mathrm{zr}}^{\mathrm{c}}}, I_t \cap I_{\mathrm{hv}}, j) + \mathcal{H}(R(X,\bar{w})_{I_{\mathrm{zr}}^{\mathrm{c}}}, I_t \setminus I_{\mathrm{hv}}).$$

Since, the weight of $I_t$ is $\leq W_{\mathrm{cn}}$, the weight of $I_t \setminus I_{\mathrm{hv}}$ is also $\leq W_{\mathrm{cn}}$. Lemma 8 implies that

$$\mathcal{H}(R(X,\bar{w})_{I_{\mathrm{zr}}^{\mathrm{c}}}, I_t \setminus I_{\mathrm{hv}}, j) \leq 4\tau^2.$$

Combining the last three equation gives,

$$\mathcal{H}(R(X,\bar{w})_{I_{\mathrm{zr}}^{\mathrm{c}}}, I_t \cap I_{\mathrm{hv}}, j) \geq 4\tau^2.$$

Next, note that

$$\mathbb{E}[Z_t \cdot \mathbb{1}_t(I_{\mathrm{hv}})] = \sum_{i \in I_t \cap I_{\mathrm{hv}}} \Pr[i^{th} \text{ row is picked}] \cdot \bar{w}^f(i) \propto \sum_{i \in I_t \cap I_{\mathrm{hv}}} \frac{\mathcal{H}(R(X,\bar{w})_{I_{\mathrm{zr}}^{\mathrm{c}}}, i, j)}{\bar{w}^f(i)} \cdot \bar{w}^f(i)$$

$$= \mathcal{H}(R(X, \bar{w})_{I^c_{zr}}, I_t \cap I_{hv}, j),$$

where we used the fact that the probability of adding row $i$ to $I_{zr}$ by Row-Deletion procedure is proportional to $\frac{\sigma_j^2 u_j^2(i)}{\bar{w}^f(i)} = \frac{\mathcal{H}(R(X,\bar{w})_{I^c_{zr}}, i, j)}{\bar{w}^f(i)}$

A similar calculation implies that

$$\mathbb{E}[Z_t \cdot \mathbb{1}_t(I^c_{hv})] \propto \mathcal{H}(R(X, \bar{w})_{I^c_{zr}}, I_t \setminus I_{hv}, j).$$

Combining the last four equations show,

$$\mathbb{E}[Z_t \cdot \mathbb{1}_t(I^c_{hv})] \leq \mathbb{E}[Z_t \cdot \mathbb{1}_t(I_{hv})].$$

Let random variable $\ell$ denote the total number of rows that are added to $I_{zr}$ before algorithm stops. Then using the optional stopping theorem for supermartingale $(Z_t \cdot \mathbb{1}_t(I^c_{hv}) - Z_t \cdot \mathbb{1}_t(I_{hv}))$ implies that when algorithm stops after putting $\ell$ rows in $I_{zr}$,

$$\mathbb{E}[\sum_{t=1}^{\ell} (Z_t \cdot \mathbb{1}_t(I^c_{hv}) - Z_t \cdot \mathbb{1}_t(I_{hv}))] \leq 0.$$

And since, the total weight of the rows in $I_{hv}$ that gets added to $I_{zr}$ is at-most the total weight of all rows in $I_{hv}$, using Lemma 8, we get

$$\sum_{t=1}^{\ell} Z_t \cdot \mathbb{1}_t(I_{hv}) \leq \mathcal{O}(k/n_{avg}).$$

Combining the above two equations bounds the expected total weight of rows that gets added to $I_{zr}$

$$\mathbb{E}[\sum_{t=1}^{\ell} Z_t] = \mathbb{E}[\sum_{t=1}^{\ell} Z_t \cdot \mathbb{1}_t(I^c_{hv})] + \mathbb{E}[\sum_{t=1}^{\ell} Z_t \cdot \mathbb{1}_t(I_{hv})] \leq 2\mathbb{E}[\sum_{t=1}^{\ell} Z_t \cdot \mathbb{1}_t(I_{hv})] \leq 2\mathcal{O}(k/n_{avg}). \quad \blacksquare$$

**Lemma 12.** *Assume that the essential property* (1) *holds. If Curated SVD is run* $\mathcal{O}(\log k)$ *times, then w.p.* $> 1 - k^{-O(1)}$, *at-least one of the runs finds* $I_{zr}$ *s.t.,*

$$\bar{w}^f(I_{zr}) \leq \mathcal{O}(k/n_{avg}).$$

*Proof.* From Markov's inequality and the previous lemma, we get

$$\Pr[\bar{w}^f(I_{zr}) \geq 5 \times \mathcal{O}(k/n_{avg}))] \leq \frac{\mathcal{O}(k/n_{avg})}{5 \times \mathcal{O}(k/n_{avg})} \leq 1/5.$$

Therefore, w.p. $\geq 4/5$, Curated SVD finds $I_{zr}$ such that,

$$\bar{w}^f(I_{zr}) \leq 5 \times \mathcal{O}(k/n_{avg}).$$

Hence, if we run Curated SVD for $\mathcal{O}(\log k)$ times, with probability $1 - k^{-O(1)}$, at-least one of the runs will find $I_{zr}$ that satisfy the above equation and Objective **(ii)**. $\blacksquare$

### C.2  Objectives (i) and (ii) implies recovery

**Lemma 13.** *Assume that essential property* (1) *holds and let* $I_{zr}$ *be any row subset satisfying objectives (i) and (ii) and let* $\widehat{M}$ *be* $(2r, \bar{w})$-*SVD of* $X_{I^c_{zr}}$, *then*

$$||M_{I^c_{cn}} - \widehat{M}||_1 \leq \mathcal{O}(k\sqrt{rn_{avg}} \log(rn_{avg})).$$

The rest of this subsection proves the lemma.

Let $Z := R(X_{I^c_{zr}}, \bar{w})^{(2r)}$, be rank $2r$ truncated SVD of regularized matrix $R(X_{I^c_{zr}}, \bar{w})$. Then, from the definition of $(2r, \bar{w})$-SVD,

$$\widehat{M} = D^{\frac{1}{2}}(\bar{w}^f) \cdot Z \cdot D^{\frac{1}{2}}(\bar{w}^b).$$

Clearly, both $Z$ and $\widehat{M}$ have rank $\leq 2r$ and their rows $I_{\text{zr}}$ are all zero.

To bound the total loss in Lemma 13, using the triangle inequality, we first decompose the loss incurred in estimating matrix $M$ by $\widehat{M}$ into three parts as follows.

$$\begin{aligned}
||M_{I_{\text{cn}}^{\text{c}}} - \widehat{M}||_1 &= ||M_{I_{\text{cn}}^{\text{c}}} - \widehat{M}_{I_{\text{zr}}^{\text{c}}}||_1 \\
&\leq ||(M - \widehat{M})_{(I_{\text{cn}} \cup I_{\text{zr}})^{\text{c}}}||_1 + ||M_{I_{\text{zr}} \setminus I_{\text{cn}}}||_1 + ||\widehat{M}_{I_{\text{cn}} \setminus I_{\text{zr}}}||_1.
\end{aligned} \tag{5}$$

Lemma 14 bounds the contribution of the first term above on the right. Intuitively, the contribution of the other two terms is small, since the weight of row subsets $I_{\text{zr}}$ and $I_{\text{cn}}$ are small. And using this observation we later bound the contribution of the last two terms.

**Lemma 14.**
$$||(M - \widehat{M})_{(I_{\text{cn}} \cup I_{\text{zr}})^{\text{c}}}||_1 \leq \mathcal{O}(k\sqrt{rn_{avg}}\log(rn_{avg})).$$

To prove Lemma 14, we use the following lemma which bounds the spectral distance between $Z_{(I_{\text{cn}} \cup I_{\text{zr}})^{\text{c}}}$ and $R(M, \bar{w})_{(I_{\text{cn}} \cup I_{\text{zr}})^{\text{c}}}$ is small.

**Lemma 15.** *Let $I_{\text{zr}} \subseteq [k]$ be a row subset that satisfy Objective-1, then*
$$||Z_{(I_{\text{cn}} \cup I_{\text{zr}})^{\text{c}}} - R(M, \bar{w})_{(I_{\text{cn}} \cup I_{\text{zr}})^{\text{c}}}|| \leq 9\tau.$$

In proving the above Lemma, we need the following auxiliary lemmas, which is stated for general matrices, and its proof appears in the Appendix G

**Lemma 16.** *Let $A = B + C$ and $A = \sum_i \sigma_i(A) u_i v_i^{\mathsf{T}}$ be the SVD decomposition of $A$. And $\sigma_{r+1}(B) \leq \beta$ and $||Cv_i|| \leq 2\beta$ for $i \in [2r]$. Then $\sigma_{2r}(A) \leq 4\beta$.*

*Proof of Lemma 15.* Using equation (3) and Theorem 6, we get,
$$\sigma_{r+1}(R(X, \bar{w})_{(I_{\text{cn}} \cup I_{\text{zr}})^{\text{c}}}) \leq \sigma_{r+1}(R(X, \bar{w})_{I_{\text{cn}}^{\text{c}}}) \leq \tau. \tag{6}$$

Observe that,
$$R(X, \bar{w})_{I_{\text{zr}}^{\text{c}}} = R(X, \bar{w})_{(I_{\text{cn}} \cup I_{\text{zr}})^{\text{c}}} + R(X, \bar{w})_{I_{\text{cn}} \setminus I_{\text{zr}}}.$$

Let $(R(X, \bar{w})_{I_{\text{zr}}^{\text{c}}})^{(2r)} = \sum_{j=1}^{2r} \sigma_j u_j v_j^{\mathsf{T}}$. Since $I_{\text{zr}}$ satisfy Objective (i) and the weight of $I_{\text{cn}}$ is at-most $\bar{w}^f(I_{\text{cn}}) \leq \epsilon k$, therefore, the condition in Objective (i) implies that
$$||R(X, \bar{w})_{I_{\text{cn}} \setminus I_{\text{zr}}} \cdot v_i|| \leq 4\tau.$$

Then applying Lemma 16, for $A = R(X, \bar{w})_{I_{\text{zr}}^{\text{c}}}$ and using the above three equations gives
$$\sigma_{2r}(R(X, \bar{w})_{I_{\text{zr}}^{\text{c}}}) \leq 8\tau.$$

Since, for $Z$ is rank-$2r$ truncated SVD of $R(X, \bar{w})_{I_{\text{zr}}^{\text{c}}}$, we have
$$||R(X, \bar{w})_{I_{\text{zr}}^{\text{c}}} - Z|| = ||(R(X, \bar{w}) - Z)_{I_{\text{zr}}^{\text{c}}}|| \leq 8\tau. \tag{7}$$

Then,
$$\begin{aligned}
&||(Z - R(M, \bar{w}))_{(I_{\text{cn}} \cup I_{\text{zr}})^{\text{c}}}|| \\
&\overset{(a)}{\leq} ||(Z - R(X, \bar{w}))_{(I_{\text{cn}} \cup I_{\text{zr}})^{\text{c}}}|| + ||(R(X, \bar{w}) - R(M, \bar{w}))_{(I_{\text{cn}} \cup I_{\text{zr}})^{\text{c}}}|| \\
&\overset{(b)}{\leq} ||(Z - R(X, \bar{w}))_{I_{\text{zr}}^{\text{c}}}|| + ||(R(N, \bar{w}))_{I_{\text{cn}}^{\text{c}}}|| \\
&\overset{(c)}{\leq} 8\tau + \tau = 9\tau,
\end{aligned}$$

here (a) uses triangle inequality, (b) uses Theorem 6 and $X - M = N$, and finally (c) uses equation (7) and essential condition (1). ∎

The next auxiliary lemma relates the $L_1$ norm and spectral norm. Its proof appears in the Appendix G.

**Lemma 17.** *For any rank-$r$ matrix $A \in \mathbb{R}^{k \times m}$ and weight vectors $w^f$ and $w^b$ with non-negative entries*
$$||D^{\frac{1}{2}}(w^f) \cdot A \cdot D^{\frac{1}{2}}(w^b)||_1 \leq \sqrt{r\left(\sum_i w^f(i)\right)\left(\sum_j w^b(j)\right)} \cdot ||A||.$$

We will also need the following result.

**Lemma 18.** *The total weight of all rows is at most $\sum_{i\in[k]} \bar{w}^f(i) \leq 2k$ and similarly the total weight of all columns is at most $\sum_{i\in[k]} \bar{w}^b(j) \leq 2k$.*

*Proof.*

$$\bar{w}^f([k]) = \sum_{i\in[k]} \bar{w}^f(i) \leq \sum_{i\in[k]} 1 + \frac{||X_{i,*}||_1}{n_{\text{avg}}} \leq k + \frac{||X||_1}{n_{\text{avg}}} \leq 2k,$$

since $n_{\text{avg}}$ is the average number of samples in each row. Similarly it can be shown for columns. ∎

Next, combining the above result we prove Lemma 14

*Proof of Lemma 14.* First note that,

$$||(M - \widehat{M})||_1 = ||M - D^{\frac{1}{2}}(\bar{w}^f) \cdot Z \cdot D^{\frac{1}{2}}(\bar{w}^b)||_1 = ||D^{\frac{1}{2}}(\bar{w}^f)(Z - R(M,\bar{w}))D^{\frac{1}{2}}(\bar{w}^b)||_1,$$

here we used $R(M,\bar{w}) = D^{-\frac{1}{2}}(\bar{w}^f) \cdot M \cdot D^{-\frac{1}{2}}(\bar{w}^b)$. As noted earlier $Z$ has the rank $\leq 2r$ and $M$ has the rank $\leq r$, therefore the rank of $(Z - R(M,\bar{w}))$ is at most $3r$. Then, using Lemma 17 and Lemma 15,

$$||D^{\frac{1}{2}}(\bar{w}^f)(Z - R(M,\bar{w}))_{(I_{\text{cn}}\cup I_{\text{zr}})^c} D^{\frac{1}{2}}(\bar{w}^b)||_1$$

$$\leq \sqrt{3r}\sqrt{(\sum_{j\in[k]} \bar{w}^b(j))(\sum_{i\in(I_{\text{cn}}\cup I_{\text{zr}})^c} \bar{w}^b(j))} \cdot 9\tau$$

$$\leq \sqrt{3r}\sqrt{(\sum_{j\in[k]} \bar{w}^b(j))(\sum_{i\in[k]} \bar{w}^b(j))} \cdot 9\tau \leq \sqrt{3r} \cdot 2k \cdot 9\tau,$$

here the last step uses Lemma 18. Combining the last two equations and using $\tau = \mathcal{O}(\sqrt{n_{\text{avg}}}\log(rn_{\text{avg}}))$ complete the proof. ∎

To bound the second term in (5), note that

$$M_{I_{\text{zr}}\backslash I_{\text{cn}}} = D^{\frac{1}{2}}(\bar{w}^f) \cdot R(M,\bar{w})_{I_{\text{zr}}\backslash I_{\text{cn}}} \cdot D^{\frac{1}{2}}(\bar{w}^b) \tag{8}$$

Applying Lemma 17,

$$||M_{I_{\text{zr}}\backslash I_{\text{cn}}}||_1 = ||D^{\frac{1}{2}}(\bar{w}^f) \cdot R(M,\bar{w})_{I_{\text{zr}}\backslash I_{\text{cn}}} \cdot D^{\frac{1}{2}}(\bar{w}^b)||_1$$

$$\leq \sqrt{r}\sqrt{(\sum_{j\in[k]} \bar{w}^b(j))(\sum_{i\in I_{\text{zr}}\backslash I_{\text{cn}}} \bar{w}^f(i))} \cdot ||R(M,\bar{w})_{I_{\text{zr}}\backslash I_{\text{cn}}}||$$

$$\leq \sqrt{r}\sqrt{(\sum_{j\in[k]} \bar{w}^b(j))(\sum_{i\in I_{\text{zr}}} \bar{w}^f(i))} \cdot ||R(M,\bar{w})_{I_{\text{cn}}^c}||$$

$$\leq \sqrt{2rk}\sqrt{\bar{w}^f(I_{\text{zr}})} \cdot ||R(M,\bar{w})_{I_{\text{cn}}^c}||. \tag{9}$$

Next,

$$||R(M,\bar{w})_{I_{\text{cn}}^c}|| \leq ||R(N,\bar{w})_{I_{\text{cn}}^c}|| + ||R(X,\bar{w})_{I_{\text{cn}}^c}||$$

$$\leq \tau + ||R(X,\bar{w})|| \leq \tau + n_{\text{avg}},$$

here we used essential property (1) and equation (4). Combining the last two equations we get

$$||M_{I_{\text{zr}}\backslash I_{\text{cn}}}||_1 \leq \mathcal{O}(\sqrt{rk \cdot \bar{w}^f(I_{\text{zr}})} \cdot (\tau + n_{\text{avg}})) \leq \mathcal{O}(\sqrt{rk} \cdot (\log(rn_{\text{avg}}) + \sqrt{n_{\text{avg}}})) \tag{10}$$

Finally, we bound the last term in (5). Recall that $\widehat{M} = D^{\frac{1}{2}}(\bar{w}^f) \cdot Z \cdot D^{\frac{1}{2}}(\bar{w}^b)$. Then

$$||\widehat{M}_{I_{\text{cn}}\backslash I_{\text{zr}}}||_1 = ||(D^{\frac{1}{2}}(\bar{w}^f) \cdot Z_{I_{\text{cn}}\backslash I_{\text{zr}}} \cdot D^{\frac{1}{2}}(\bar{w}^b)||_1.$$

Using the fact that $Z$ is truncated SVD of $R(X_{I_{\text{zr}}^c},\bar{w})$ and equation (4) we get

$$||Z_{I_{\text{zr}}^c}|| \leq ||Z|| \leq ||R(X_{I_{\text{zr}}^c},\bar{w})|| \leq n_{\text{avg}}.$$

Therefore,

$$||Z_{I_{\text{cn}}\backslash I_{\text{zr}}}|| \leq n_{\text{avg}}.$$

Then applying Lemma 17,

$$||D^{\frac{1}{2}}(\bar{w}^f) \cdot Z_{I_{\text{cn}} \setminus I_{\text{zr}}} \cdot D^{\frac{1}{2}}(\bar{w}^b)||_1 \leq \sqrt{2r}\sqrt{(\sum_{j \in [k]} \bar{w}^b(j))(\sum_{i \in I_{\text{cn}} \setminus I_{\text{zr}}} \bar{w}^f(i))} \cdot n_{\text{avg}}$$

$$\leq \sqrt{2r}\sqrt{2k(\sum_{i \in I_{\text{cn}}} \bar{w}^f(i))} \cdot n_{\text{avg}}$$

$$\leq 2\sqrt{rkW_{\text{cn}}} \cdot n_{\text{avg}} = \mathcal{O}(k/\sqrt{r}),$$

here in the last step we used $W_{\text{cn}} = \mathcal{O}(k/(rn_{\text{avg}})^2)$.

By combining equation (5), Lemma 14, equation (10) and the above equations we get

$$||M_{I_{\text{cn}}} - \widehat{M}||_1 \leq \mathcal{O}(k\sqrt{rn_{\text{avg}}} \log(rn_{\text{avg}})) + \mathcal{O}(k\sqrt{r} \cdot (\log(rn_{\text{avg}}) + \sqrt{n_{\text{avg}}})) + \mathcal{O}(k/\sqrt{r})$$

$$\leq \mathcal{O}(k\sqrt{rn_{\text{avg}}} \log(rn_{\text{avg}})).$$

This completes the proof of the lemma.

## D Proof of Theorem 2

Theorem 2 follows immediately from the following theorem.

**Theorem 19.** *Curated SVD runs in polynomial time, and for every $k$, $r$, $\epsilon > 0$, $M \in \mathbb{R}_r^{k \times k}$, and $X \sim M$, returns an estimate $M^{cur}(X)$ s.t. with probability $\geq 1 - k^{-2}$,*

$$L(M^{cur}(X)) = \frac{||M - M^{cur}(X)||_1}{||M||_1} \leq \mathcal{O}(\sqrt{\frac{kr}{||M||_1}} \log(\frac{r||M||_1}{k})).$$

*Proof.* Section 4 showed that Curated-SVD always achieves Objective (i). Using the spectral concentration bound in Theorem 3 it also showed that the essential property required for Curated SVD holds with probability $\geq 1 - 6/k^3$.

Lemma 12 showed that if essential property hold and Curated SVD is repeated $\mathcal{O}(\log k)$ times, on the same samples, then w.p. $> 1 - k^{-O(1)}$, at-least one of the runs find $I_{\text{zr}}$ that achieves Objective (ii).

Finally, when essential property holds, and Curated SVD achieves both the objectives then Lemma 13 showed

$$||M_{I_{\text{cn}}^c} - \widehat{M}||_1 \leq \mathcal{O}(k\sqrt{rn_{\text{avg}}} \log(rn_{\text{avg}})). \tag{11}$$

Note that

$$||M - \widehat{M}||_1 \leq ||M_{I_{\text{cn}}^c} - \widehat{M}||_1 + ||M_{I_{\text{cn}}}||_1.$$

Therefore, to prove the theorem the only thing remains is to bound the last term.

To bound the last term we use the following lemma. The proof of the Lemma appears in Section E.1.

The lemma upper bounds the sum of the absolute difference between the expected and observed samples in each row of $X$.

**Lemma 20.** *With probability $\geq 1 - 3k^{-3}$,*

$$\sum_i |\sum_j N_{i,j}| = \mathcal{O}(k\sqrt{n_{avg}}).$$

Then

$$||M_{I_{\text{cn}}}||_1 = \sum_{i \in I_{\text{cn}}} \sum_{j \in [k]} M_{i,j}$$

$$\leq \sum_{i \in I_{\text{cn}}} \sum_{j \in [k]} X_{i,j} + \left| \sum_{i \in I_{\text{cn}}} \sum_{j \in [k]} (M_{i,j} - X_{i,j}) \right|$$

$$\leq \sum_{i \in I_{\text{cn}}} ||X_{i,*}|| + \sum_{i \in I_{\text{cn}}} \left| \sum_{j \in [k]} N_{i,j} \right|$$

$$\leq \sum_{i \in I_{\text{cn}}} n_{\text{avg}} \cdot \bar{w}^f(i) + \mathcal{O}(k\sqrt{n_{\text{avg}}})$$
$$\leq \mathcal{O}(k\sqrt{n_{\text{avg}}}),$$

here the second last step uses $||X_{i,*}|| \leq n_{\text{avg}} \cdot \bar{w}^f(i)$ and the previous Lemma and the last step uses $\sum_{i \in I_{\text{cn}}} \bar{w}^f(i) \leq W_{\text{cn}}$. Combining this with (11) and letting $M^{\text{cur}}(X) = \widehat{M}$ gives

$$||M - M^{\text{cur}}(X)||_1 \leq \mathcal{O}(k\sqrt{rn_{\text{avg}}} \log(rn_{\text{avg}})).$$

Finally, dividing the both sides by $||M||_1$ and using $n_{\text{avg}} = ||M||_1/k$ in the above equation completes the proof. ∎

Using $||M||_1 = \frac{kr}{\epsilon^2} \cdot \log^2 \frac{r}{\epsilon}$ in the above theorem gives Theorem 2.

The next subsection gives the implication of the above result for collaborative filtering.

### D.1 Collaborative filtering

For the same general bounded noise model [BCLS17] derived the mean square error $\sum_{i,j}(F_{i,j} - \hat{F}_{i,j})^2/k^2 = \mathcal{O}(r^2/(pk)^{2/5})$. They assume that the mean matrix $F$ is generated by a Lipschitz latent variable model. Here we show that Curated SVD achieves a better accuracy, in a stronger norm, and without the additional Lipschitz assumption on $F$.

Note that since $0 \leq F_{i,j} \leq 1$, $|F_{i,j} - \hat{F}_{i,j}| \geq |F_{i,j} - \hat{F}_{i,j}|^2$. Therefore, $L_1$ error $\sum_{i,j} |F_{i,j} - \hat{F}_{i,j}|/k^2$ upper bounds mean squared error.

Recall that in collaborative filtering model $M_{i,j} = pF_{i,j}$. Since the sampling probability $p$ can be estimated to very good accuracy hence without loose of generality assume that $p$ is known. Note that $||M||_1 = p||F||_1 = pk^2 F_{i,j}^{\text{avg}}$, where $F_{i,j}^{\text{avg}} = ||F||_1/k^2 \leq 1$ as $\forall i, j$, $F_{i,j} \leq 1$. We let $F^{\text{cur}}(X) = M^{\text{cur}}(X)/p$.

Then using Theorem 19 for this model implies:

$$\frac{||M - M^{\text{cur}}(X)||_1}{||M||_1} = \frac{p||F - F^{\text{cur}}(X)||_1}{pk^2 F_{i,j}^{\text{avg}}} \leq \mathcal{O}\left(\sqrt{\frac{kr}{pk^2 F_{i,j}^{\text{avg}}}} \log(\frac{r}{k}pk^2 F_{i,j}^{\text{avg}})\right).$$

Therefore,

$$\frac{||F - F^{\text{cur}}(X)||_1}{k^2} \leq \mathcal{O}\left(\sqrt{\frac{rF_{i,j}^{\text{avg}}}{pk}} \log(r \cdot pkF_{i,j}^{\text{avg}})\right) \leq \mathcal{O}\left(\sqrt{\frac{r}{pk}} \log(rpk)\right).$$

We get the following Corollary.

**Corollary 21.** *Curated SVD runs in polynomial time, and for every $k$, $r$, $\epsilon > 0$, sampling probability $p$, $F \in \mathbb{R}_r^{k \times k}$, $F_{i,j} \in [0, 1]$ and $X \sim pF$, returns an estimate $F^{cur}(X)$ s.t. with probability $\geq 1 - k^{-2}$,*

$$\frac{||F - F^{cur}(X)||_1}{k^2} \leq \mathcal{O}\left(\sqrt{\frac{r}{pk}} \log(rpk)\right).$$

Note that the above bound on the $L_1$ error norm is strictly better than the previous bound on the mean square error, and, as we showed, MSE is also a weaker error norm than $L_1$ for this setting.

## E   Properties of the Noise matrix

Here we give the proof of Theorem 3. We in fact prove a somewhat more general version of the theorem. Accordingly, we define the generalisation of the weights $\tilde{w}$ and $\bar{w}$ defined in the paper. For any $\Lambda > 0$, define weights $\tilde{w}_\Lambda := (\tilde{w}_\Lambda^f, \tilde{w}_\Lambda^b)$ such that,

$$\tilde{w}_\Lambda^f(i) := \max\{1, \frac{||M_{i,*}||_1}{\Lambda}\} \text{ and } \tilde{w}_\Lambda^b(j) := \max\{1, \frac{||M_{*,j}||_1}{\Lambda}\}.$$

And similarly define $\bar{w}_\Lambda = (\bar{w}_\Lambda^f, \bar{w}_\Lambda^b)$ such that,

$$\bar{w}_\Lambda^f(i) := \max\{1, \tfrac{||X_{i,*}||_1}{\Lambda}\} \text{ and } \bar{w}_\Lambda^b(j) := \max\{1, \tfrac{||X_{*,j}||_1}{\Lambda}\}.$$

We obtain the results for the regularization weights $\tilde{w}_\Lambda^f(i)$ and $\bar{w}_\Lambda^f(i)$. Note that the regularization weights $\tilde{w}$ and $\bar{w}$, used in the main paper, are a special case of these regularization weights $\tilde{w}_\Lambda^f(i)$ and $\bar{w}_\Lambda^f(i)$ for $\Lambda = n_{\mathrm{avg}}$.

The next theorem is a generalisation of Theorem 3. This theorem bounds the spectral norm of the noise matrix regularized by weights $\bar{w}_\Lambda$, and Theorem 3 can be obtained as a special case of this theorem for $\Lambda = n_{\mathrm{avg}}$.

**Theorem 22.** *For $X \sim M$, any $\Lambda > 0$, $\epsilon \geq \frac{1}{\Lambda} \max\left(\frac{\log^4 k}{k}, \exp^{-\frac{\Lambda}{8}}\right)$, with probability $\geq 1 - 6k^{-3}$, there is a row subset $I_{\mathrm{cn}} \subseteq [k]$ of possibly contaminated rows such that $\sum_{i \in I} \bar{w}_\Lambda^f(i) \leq \epsilon k$ and*

$$||R(N, \bar{w}_\Lambda)_{I_{\mathrm{cn}}^{\mathrm{c}}}|| \leq \mathcal{O}\big(\sqrt{\Lambda} \cdot \log \tfrac{2}{\epsilon}\big).$$

To prove the above theorem we first establish the bound on $\ell_\infty \to \ell_2$ norm of submatrices of the regularized noise matrix and use a known result, referred as Grothendieck-Pietsch factorization, to relate this norm to spectral norm. Next we define $\ell_\infty \to \ell_2$ norm and state Grothendieck-Pietsch factorization.

The $\ell_\infty \to \ell_2$ norm of a matrix $A \in \mathbb{R}^{k \times m}$ is

$$||A||_{\infty \to 2} := \max_{||v||_\infty = 1} ||Av||_2 = \max_{v \in \{-1,1\}^m} ||Av||_2.$$

Since the vector $v$ in this definition takes value in the finite set ($\{-1, 1\}^m$), standard probabilistic techniques are better suited for bounding the $\ell_\infty \to \ell_2$ norm than for bounding the spectral norm directly. In turn, Grothendieck-Pietsch factorization helps us relate $\ell_\infty \to \ell_2$ and spectral norm.

**Theorem 23.** (Grothendieck-Pietsch factorization)
*For any $A \in \mathbb{R}^{k \times m}$ there is a vector $\mu = (\mu(1), ..., \mu(m))$ with $\mu(j) \geq 0$ and $\sum_j \mu(j) = 1$ such that*

$$||A \cdot D^{-\frac{1}{2}}(\mu)|| \leq \sqrt{\frac{\pi}{2}} \cdot ||A||_{\infty \to 2}.$$

The above result can be obtained by combining Little Grothendieck Theorem and Pietsch Factorization, and has appeared in [LT13] (Proposition 15.11) and [LLV17] (Theorem 3.1).

To prove the above theorem, therefore, we first bound the $\ell_\infty \to \ell_2$ norm of submatrices of $D^{-\frac{1}{2}}(\bar{w}_\Lambda^f) \cdot N$. The proof of the lemma is based on standard use of the probabilistic methods. Due to the symmetry, a similar bound will hold on $\ell_\infty \to \ell_2$ norm of submatrices of $(N \cdot D^{-\frac{1}{2}}(\bar{w}_\Lambda^b))^\mathsf{T} = D^{-\frac{1}{2}}(\bar{w}_\Lambda^b) \cdot N^\mathsf{T}$.

**Lemma 24.** ($\ell_\infty \to \ell_2$ *concentration*) *With probability $\geq 1 - 3k^{-3}$, for every* $\max(\frac{\log^4 k}{\Lambda}, \frac{k \exp^{-\frac{\Lambda}{8}}}{\Lambda}) \leq \ell \leq k$, *and every $I, J \subseteq [k]$ of size $\ell$,*

$$||(D^{-\frac{1}{2}}(\bar{w}_\Lambda^f) \cdot N)_{I \times J}||_{\infty \to 2} \leq \mathcal{O}(\sqrt{\Lambda \ell \log(ek/\ell)}).$$

*Proof.*

$$||(D^{-\frac{1}{2}}(\bar{w}_\Lambda^f) \cdot N)_{I \times J}||_{\infty \to 2}^2 = \max_{v \in \{-1,1\}^\ell} ||(D^{-\frac{1}{2}}(\bar{w}_\Lambda^f) \cdot N)_{I \times J} \cdot v||^2$$

$$= \max_{v \in \{-1,1\}^\ell} \sum_{i \in I} \left(\sum_{j \in J} \frac{N_{i,j}}{\sqrt{\bar{w}_\Lambda^f(i)}} v(j)\right)^2$$

$$= \max_{v \in \{-1,1\}^\ell} \sum_{i \in I} \frac{Z_i(v)^2}{\bar{w}_\Lambda^f(i)} = \max_{v \in \{-1,1\}^\ell} \sum_{i \in I} \widehat{Z}_i(v)^2,$$

where

$$Z_i(v) = \sum_{j \in J} N_{i,j} v(j) \quad \text{and} \quad \widehat{Z}_i(v) = \frac{Z_i(v)}{\sqrt{\bar{w}_\Lambda^f(i)}}.$$

For a fix $v$, $Z_i(v)$ is the sum of independent zero-mean random variables, the following bound follows from Bernstein's inequality, for any $t > 0$

$$\Pr(|Z_i(v)| > t) \le 2\exp\left(\frac{-t^2/2}{||M_{i,*}||_1 + t/3}\right) \le 2\exp\left(\frac{-t^2/2}{\Lambda \tilde{w}_\Lambda^f(i) + t/3}\right). \qquad (12)$$

Observe that

$$|Z_i(v)| \le \sum_{j \in J}|N_{i,j}| \le ||N_{i,*}||_1 \le ||X_{i,*}||_1 + ||M_{i,*}||_1 \le (\bar{w}_\Lambda^f(i) + \tilde{w}_\Lambda^f(i))\Lambda. \qquad (13)$$

Based on the values of $\tilde{w}_\Lambda^f(i)$ and $\bar{w}_\Lambda^f(i)$ we divide the rows into two categories, and let random variable $T_i$ denote the category of row $i$,

$$T_i := \begin{cases} 1 & \text{if } \bar{w}_\Lambda^f(i) \ge \frac{\tilde{w}_\Lambda^f(i)}{2}, \\ 2 & \text{else .} \end{cases}$$

Let $\xi_i := \mathbb{1}_{\{T_i=1\}}$ be the indicator random variable corresponding to the event that row $i$ is in the first category. Hence, $\bar{\xi}_i := 1 - \xi_i = \mathbb{1}_{\{T_i=2\}}$. Then

$$\sum_{i \in I} \widehat{Z}_i(v)^2 = \sum_{i \in I}(\xi_i + \bar{\xi}_i)\widehat{Z}_i(v)^2.$$

Next, we bound the contribution of the rows in each categories to the above term.

1. $T_i = 1:\ \bar{w}_\Lambda^f(i) \ge \frac{\tilde{w}_\Lambda^f(i)}{2}$.

   To bound the contribution of the rows in category 1, we show for any given $v$, $\xi_i\widehat{Z}_i(v)$ are sub-Gaussian random variables with sub-Gaussian norm $\mathcal{O}(\sqrt{\Lambda})$. Then we bound sum of their squares, which are sub-exponential random variable, by applying Bernstien's concentration bound. We first prove that $\xi_i\widehat{Z}_i(v)$ are sub-Gaussian.

   From equation (13) and the definition of category 1, we get

   $$|\xi_i Z_i(v)| \le |Z_i(v)| \le (\bar{w}_\Lambda^f(i) + \tilde{w}_\Lambda^f(i))\Lambda \le (\bar{w}_\Lambda^f(i) + 2\bar{w}_\Lambda^f(i))\Lambda = 3\bar{w}_\Lambda^f(i)\Lambda,$$

   hence

   $$\bar{w}_\Lambda^f(i) \ge \frac{|\xi_i Z_i(v)|}{\Lambda}.$$

   Then

   $$|\xi_i\widehat{Z}_i(v)| = \frac{|\xi_i Z_i(v)|}{\sqrt{\bar{w}_\Lambda^f(i)}} \le \min\left\{|\sqrt{\Lambda Z_i(v)}|, \frac{|Z_i(v)|}{\sqrt{\tilde{w}_\Lambda^f(i)/2}}\right\},$$

   here we used the previous equation, the fact that $\xi_i \le 1$, and $\bar{w}_\Lambda^f(i) \ge \tilde{w}_\Lambda^f(i)/2$, which follows from the definition of the category 1. Using equation (12) we get

   $$\Pr\left(\frac{|Z_i(v)|}{\sqrt{\tilde{w}_\Lambda^f(i)/2}} > t\right) = \Pr\left(|Z_i(v)| > t\sqrt{\frac{\tilde{w}_\Lambda^f(i)}{2}}\right) \le 2\exp\left(\frac{-t^2\tilde{w}_\Lambda^f(i)/4}{\Lambda \tilde{w}_\Lambda^f(i) + \frac{t}{3}\cdot\sqrt{\frac{\tilde{w}_\Lambda^f(i)}{2}}}\right).$$

   For $t \le 3\Lambda\sqrt{\frac{\tilde{w}_\Lambda^f(i)}{2}}$, the above equation gives the following bound

   $$\Pr\left(\frac{|Z_i(v)|}{\sqrt{\tilde{w}_\Lambda^f(i)/2}} > t\right) \le 2\exp\left(\frac{-t^2\tilde{w}_\Lambda^f(i)/4}{\Lambda \tilde{w}_\Lambda^f(i) + \Lambda \cdot \frac{\tilde{w}_\Lambda^f(i)}{2}}\right) = 2\exp\left(\frac{-t^2}{6\Lambda}\right). \qquad (14)$$

   And, similarly

   $$\Pr(|\sqrt{Z_i(v)\Lambda}| > t) = \Pr\left(|Z_i(v)| > \frac{t^2}{\Lambda}\right) \le 2\exp\left(\frac{-\frac{t^4}{2\Lambda^2}}{\Lambda \tilde{w}_\Lambda^f(i) + \frac{t^2}{3\Lambda}}\right).$$

For $t \geq 3\Lambda\sqrt{\frac{\tilde{w}_\Lambda^f(i)}{2}}$, the above equation give the following bound

$$\Pr(|\sqrt{Z_i(v)\Lambda}| > t) \leq 2\exp\left(\frac{-\frac{t^4}{2\Lambda^2}}{\frac{2t^2}{9\Lambda} + \frac{t^2}{3\Lambda}}\right) = 2\exp\left(\frac{-t^2}{10\Lambda/9}\right).$$

Combining equation (14) and the above equation we get

$$\Pr(|\xi_i\widehat{Z}_i(v)| > t) \leq 2\exp\left(\frac{-t^2}{6\Lambda}\right).$$

This shows that $\xi_i\widehat{Z}_i(v)$ is sub-Gaussian with sub-Gaussian norm $\leq \sqrt{3\Lambda}$. Therefore, $\xi_i\widehat{Z}_i^2(v)$ is sub-exponential with sub-exponential norm $\leq 3\Lambda$. From Bernstein's inequality, for all $\varepsilon \geq 1$,

$$\Pr\left(\sum_{i \in I} \xi_i\widehat{Z}_i^2(v) > \varepsilon\ell\Lambda\right) \leq 2\exp^{-c\varepsilon\ell}.$$

Choosing $\varepsilon = (14/c)\log(ek/\ell)$, bounds the above probability by $(ek/\ell)^{-7\ell}$. Taking the union bound over all possible $\ell, v, I$, and $J$, in above equation we get $\sum_{i \in I} \xi_i\widehat{Z}_i^2(v) \leq \varepsilon\ell\Lambda$ with probability at least

$$1 - \sum_{\ell=1}^{k} 2^\ell \binom{k}{\ell}^2 \left(\frac{ek}{\ell}\right)^{-7\ell} \geq 1 - k^{-3}. \tag{15}$$

2. $T_i = 2: \ \bar{w}_\Lambda^f(i) < \frac{\tilde{w}_\Lambda^f(i)}{2}$.
   Note that,

$$|\bar{\xi}_i\widehat{Z}_i(v)| \leq |\bar{\xi}_i Z_i(v)| \overset{(a)}{\leq} \frac{3}{2}\tilde{w}_\Lambda^f(i)\Lambda,$$

where (a) uses equation (13). Then,

$$\sum_{i \in I} \bar{\xi}_i\widehat{Z}_i(v)^2 \leq \sum_{i \in [k]} \bar{\xi}_i\widehat{Z}_i(v)^2 \leq \left(\frac{3}{2}\right)^2 \sum_{i \in [k]} \bar{\xi}_i\left(\tilde{w}_\Lambda^f(i)\Lambda\right)^2. \tag{16}$$

We bound the above term by showing:

**Claim:** With probability $\geq 1 - 2/k^3$,

$$\sum_{i \in I} \bar{\xi}_i\left(\tilde{w}_\Lambda^f(i)\right)^2 \leq \mathcal{O}\left(\frac{\log^4 k + ke^{-\frac{\Lambda}{8}}}{\Lambda^2}\right).$$

**Proof of the claim:** Next, once again divide the rows into categories based on the expected count, and let $S_i$ denote the category of row $i$,

$$S_i := \begin{cases} 0 & \text{if } \tilde{w}_\Lambda^f(i) \leq 2 \\ j \quad \text{for } j \geq 1, & \text{if } \tilde{w}_\Lambda^f(i) \in (2^j, 2^{j+1}]. \end{cases}$$

For a given $M$, category $S_i$ of row $i$ is determined and is not a random variable unlike $T_i$.

Note that

$$\{\tilde{w}_\Lambda^f(i) > 2\bar{w}_\Lambda^f(i)\} \equiv \{\max\{1, \tfrac{||M_{i,*}||_1}{\Lambda}\} > \max\{2, \tfrac{2||X_{i,*}||_1}{\Lambda}\}\} \equiv \{||M_{i,*}||_1 > \max\{2\Lambda, 2||X_{i,*}||_1\}\}.$$

Next,

$$\Pr\left(2||X_{i,*}||_1 \leq ||M_{i,*}||_1\right) = \Pr\left(||X_{i,*}||_1 - ||M_{i,*}||_1 \leq -\frac{||M_{i,*}||_1}{2}\right)$$

$$= \Pr\left(\sum_{j \in [n]} (X_{i,j} - M_{i,j}) \leq -\frac{\sum_{j \in [n]} M_{i,j}}{2}\right) \leq e^{-\frac{||M_{i,*}||_1}{8}},$$

here we used $\mathbb{E}[X_{i,j}] = M_{i,j}$ and Chernoff bound. Therefore,

$$\Pr[\tilde{w}_\Lambda^f(i) > 2\bar{w}_\Lambda^f(i)] \leq \begin{cases} 0, & \text{if } ||M_{i,*}||_1 \leq 2\Lambda, \\ e^{-\frac{||M_{i,*}||_1}{8}} = e^{-\frac{\Lambda\tilde{w}_\Lambda^f(i)}{8}}, & \text{if } ||M_{i,*}||_1 > 2\Lambda. \end{cases}$$

Then from the definitions of $S_i$ and $\tilde{w}_\Lambda^f(i)$, it follows that

$$\Pr[\bar{\xi}_i = 1] = \Pr[\tilde{w}_\Lambda^f(i) > 2\bar{w}_\Lambda^f(i)] \leq \begin{cases} 0, & \text{if } S_i = 0, \\ e^{-\frac{\Lambda\tilde{w}_\Lambda^f(i)}{8}} \leq e^{-\frac{2^{S_i}\Lambda}{8}}, & \text{if } S_i \geq 1. \end{cases} \tag{17}$$

Using the Chernoff bound, for any $j \geq 1$,

$$\Pr\left( \sum_{i:S_i=j} \bar{\xi}_i \geq 15 \log k + 2 \sum_{i:S_i=j} \mathbb{E}[\bar{\xi}_i] \right) \leq k^{-5}. \tag{18}$$

Let $\tau = \lfloor \log_2 \frac{16\ln k}{\Lambda} \rfloor$. For a row $i$ such that $S_i > \tau$, using (17) we get

$$\Pr[\bar{\xi}_i = 1] \leq k^{-4}.$$

Therefore, with probability $\geq 1 - \frac{\{i:S_i>\tau\}}{k^5} \geq 1 - \frac{1}{k^3}$

$$\sum_{i:S_i>\tau} \bar{\xi}_i = 0. \tag{19}$$

Then

$$\sum_{i\in[k]} \bar{\xi}_i \left(\tilde{w}_\Lambda^f(i)\right)^2 \overset{(a)}{=} \sum_{j\geq1} \sum_{i:S_i=j} \bar{\xi}_i \left(\tilde{w}_\Lambda^f(i)\right)^2$$

$$\overset{(b)}{=} \sum_{j=1}^{\tau} \sum_{i:S_i=j} \mathbb{1}_{\{T_i=2\}} \left(\tilde{w}_\Lambda^f(i)\right)^2 \leq \sum_{j=1}^{\tau} \max_{i:S_i=j}\{(\tilde{w}_\Lambda^f(i))^2\} \sum_{i:S_i=j} \bar{\xi}_i$$

$$\overset{(c)}{\leq} \sum_{j=1}^{\tau} \max_{i:S_i=j}\{(\tilde{w}_\Lambda^f(i))^2\}\left(15 \log k + 2 \sum_{i:S_i=j} \mathbb{E}[\bar{\xi}_i]\right)$$

$$\leq 15\tau \log k \max_{i:S_i\leq\tau}\{(\tilde{w}_\Lambda^f(i))^2\} + \sum_{j=1}^{\tau} \max_{i:S_i=j}\{(\tilde{w}_\Lambda^f(i))^2\} \max_{i:S_i=j}\{\mathbb{E}[\bar{\xi}_i]\}\left(2|\{i:S_i=j\}|\right)$$

$$\overset{(d)}{\leq} 15\tau \log k \left(2^{\tau+1}\right)^2 + \sum_{j=1}^{\tau} \left(|\{i:S_i=j\}|\right) \times \left(2^{2j+2} \cdot e^{-\frac{2^j\Lambda}{8}}\right)$$

$$\leq \mathcal{O}(\frac{\log^4 k}{\Lambda^2}) + \mathcal{O}\left(\max_{j\geq1}\left\{\left(2^j\Lambda\right)^2 e^{-\frac{2^j\Lambda}{8}}\right\} \cdot \Lambda^{-2} \cdot \sum_{j=1}^{\tau} |\{i:S_i=j\}|\right)$$

$$\overset{(e)}{\leq} \mathcal{O}(\frac{\log^4 k}{\Lambda^2}) + \mathcal{O}\left(\max_{j\geq1}\left\{e^{-\frac{2^j\Lambda}{16}}\right\} \cdot \frac{k}{\Lambda^2}\right)$$

$$\leq \mathcal{O}\left(\frac{\log^4 k + ke^{-\frac{\Lambda}{8}}}{\Lambda^2}\right),$$

with probability $1 - 1/k^3 - \tau/k^5 \geq 1 - 2/k^3$. Here (a) follows since $S_i = 1$ implies $\bar{\xi}_i = 0$, (b) follows from equation (19), (c) uses equation (18), (d) follows from the definition of $S_i$ and equation (17), and (e) follows as total number of rows are $k$ and $\frac{x^2 \exp(-x/8)}{\exp(-x/16)}$ is bounded for $x > 0$. This completes the proof of the claim.

Combining the Claim and (16) gives the following bound on the contribution of rows in $T_i = 2$,

$$\sum_{i\in I} \bar{\xi}_i \hat{Z}_i^2 \leq \mathcal{O}(\log^4 k + ke^{-\frac{\Lambda}{8}}),$$

with probability $\geq 1 - 2/k^3$.

Combining this bound and the bound in (15),

$$\sum_{i \in I} \widehat{Z}_i^2 = \sum_{i \in I} \bar{\xi}_i \widehat{Z}_i^2 + \sum_{i \in I} \xi_i \widehat{Z}_i^2 \leq \mathcal{O}(\log(ek/\ell)\ell\Lambda + \log^4 k + k \exp^{-\frac{\Lambda}{8}}),$$

with probability $\geq 1 - 3k^{-3}$. Noting that $\ell\Lambda \geq \max(\log^4 k, \ k \exp^{-\frac{\Lambda}{8}})$, completes the proof. ∎

The next lemma combines the bound obtained on $\ell_\infty \to \ell_2$ norm in the above lemma and Grothendieck-Piesch factorization to obtain bound on the spectral norm.

**Lemma 25.** *With probability* $\geq 1 - 3k^{-3}$, *for any* $\max(\frac{\log^4 k}{\Lambda}, \ \frac{k \exp^{-\frac{\Lambda}{8}}}{\Lambda}) \leq \ell \leq k$, *and any* $I, J \subseteq [k]$ *of size* $|I|, |J| = \ell$, *there exists a subset* $J' \subseteq J$ *such that* $\sum_{j \in J'} \bar{w}_\Lambda^b(j) \leq \ell/2$ *and*

$$\|(R(N, \bar{w}_\Lambda))_{I \times J'}\| \leq c\sqrt{\Lambda \log(ek/\ell)}.$$

*Proof.* Applying Grothendieck-Piesch factorization in Theorem 23 on matrix $(D^{-\frac{1}{2}}(\bar{w}_\Lambda^f)N)_{I \times J}$, implies that there is a vector $\mu = (\mu(1), ..., \mu(m))$ with $\mu(j) \geq 0$ and $\sum_j \mu(j) = 1$ such that

$$\|(D^{-\frac{1}{2}}(\bar{w}_\Lambda^f) \cdot N)_{I \times J} \cdot D^{-1/2}(\mu)\| \leq \sqrt{\frac{\pi}{2}} \|(D^{-\frac{1}{2}}(\bar{w}_\Lambda^f) \cdot N)_{I \times J}\|_{\infty \to 2}$$

Then,

$$\|(D^{-\frac{1}{2}}(\bar{w}_\Lambda^f) \cdot N)_{I \times J} \cdot D^{-\frac{1}{2}}(\mu)\| = \|(D^{-\frac{1}{2}}(\bar{w}_\Lambda^f) \cdot N)_{I \times J} \cdot D^{-\frac{1}{2}}(\mu) \cdot D^{-\frac{1}{2}}(\bar{w}_\Lambda^b) \cdot D^{\frac{1}{2}}(\bar{w}_\Lambda^b)\|$$

$$= \|(D^{-\frac{1}{2}}(\bar{w}_\Lambda^f) \cdot N \cdot D^{-\frac{1}{2}}(\bar{w}_\Lambda^b) \cdot D^{\frac{1}{2}}(\bar{w}_\Lambda^b \circ \frac{1}{\mu}))_{I \times J}\|$$

$$= \|(R(N, \bar{w}_\Lambda) \cdot D^{\frac{1}{2}}(\bar{w}_\Lambda^b \circ \frac{1}{\mu}))_{I \times J}\|,$$

here $\bar{w}_\Lambda^b \circ \frac{1}{\mu} := (\frac{\bar{w}_\Lambda^b(1)}{\mu(1)}, ..., \frac{\bar{w}_\Lambda^b(m)}{\mu(m)})$. Let

$$J' := \{j \in J : \frac{\bar{w}_\Lambda^b(j)}{\mu(j)} \geq \frac{\ell}{2}\}$$

and $\bar{J} = J \setminus J'$. Then

$$\|(R(N, \bar{w}_\Lambda) \cdot D^{\frac{1}{2}}(\bar{w}_\Lambda^b \circ \frac{1}{\mu}))_{I \times J}\| \geq \|(R(N, \bar{w}_\Lambda) \cdot D^{\frac{1}{2}}(\bar{w}_\Lambda^b \circ \frac{1}{\mu}))_{I \times J'}\| \geq \sqrt{\frac{\ell}{2}} \|(R(N, \bar{w}_\Lambda))_{I \times J'}\|$$

here the last step follows from the definition of $J'$. Therefore,

$$\|(R(N, \bar{w}_\Lambda))_{I \times J'}\| \leq \sqrt{\frac{\pi}{\ell}} \|(D^{-\frac{1}{2}}(\bar{w}_\Lambda^f) \cdot N)_{I \times J}\|_{\infty \to 2} \leq c\sqrt{\Lambda \log(ek/\ell)}.$$

Next, we bound the weight of the columns that are excluded from $J'$.

$$\sum_{j \in \bar{J}} \bar{w}_\Lambda^b(j) \leq \frac{\ell}{2} \sum_{j \in \bar{J}} \mu(j) \leq \frac{\ell}{2} \sum_{j \in [k]} \mu(j) = \frac{\ell}{2}.$$

∎

Applying the above lemma on $(R(N, \bar{w}_\Lambda))^\mathsf{T}$, in place of $(R(N, \bar{w}_\Lambda)))$, from the symmetry we get:

**Lemma 26.** *With probability* $\geq 1 - 3k^{-3}$, *for any* $\max(\frac{\log^4 k}{\Lambda}, \ \frac{k \exp^{-\frac{\Lambda}{8}}}{\Lambda}) \leq \ell \leq k$, *and any* $I, J \subseteq [k]$ *of size* $|I|, |J| = \ell$, *there exists a subset* $I' \subseteq I$ *such that* $\sum_{i \in I'} \bar{w}_\Lambda^f(i) \leq \ell/2$ *and*

$$\|(R(N, \bar{w}_\Lambda))_{I' \times J}\| \leq c\sqrt{\Lambda \log(ek/\ell)}.$$

The next lemma bounds the norm of a matrix using the norm of its sub-matrices. Incorporating the above bound on the norm of submatrices, this will complete the proof of theorem 3.

**Lemma 27.** *Let $A \in \mathbb{R}^{k \times m}$ and $I_1, I_2, I_3, ..., I_t$ be $t$ disjoint subsets of $[k]$ such that $\cup_{j=1}^{t} I_j = [k]$. Then $||A|| \leq \sqrt{\sum_{j=1}^{t} ||A_{I_j}||^2}$.*

Proof of the above lemma is given in Appendix G.

*Proof of theorem 22:* Main component in the proof is Lemma 25 and Lemma 26. We need to apply these lemmas in multiple rounds.

For round $j$, we apply these Lemmas for some, $\ell = \ell_j$ and $I = I_j$, and $J = J_j$ such that $|I_j|, |J_j| \leq k/2^{j-1}$, where $\ell_j$, $I_j$ and $J_j$ are defined later.

First applying Lemma 26 we get, a subset $I'_j \subseteq I_j$ such that,

$$||\big(R(N, \bar{w}_\Lambda)\big)_{I'_j \times J_j}|| \leq \mathcal{O}(\sqrt{j \cdot \Lambda}), \tag{20}$$

and the weight of the excluded rows $I_j \setminus I'_j$ is at most $\sum_{i \in I_j \setminus I'_j} \bar{w}^f_\Lambda(i) \leq k/2^j$. Since weight of each row is at-least 1, this implies that the number of rows excluded in round $j$ are also at most $|I_j \setminus I'_j| \leq k/2^j$.

Similarly, applying Lemma 25 we get, a subset $J'_j \subseteq J_j$ such that $\sum_{i \in J_j \setminus J'_j} \bar{w}^b_\Lambda(i) \leq k/2^j$,

$$||\big(R(N, \bar{w}_\Lambda)\big)_{I_j \times J'_j}|| \leq \mathcal{O}(\sqrt{j \cdot \Lambda}). \tag{21}$$

and $\sum_{i \in J_j \setminus J'_j} \bar{w}^b_\Lambda(i) \leq k/2^j$ and $|J_j \setminus J'_j| \leq k/2^j$. Since zeroing out rows from a matrix reduces the spectral norm, the above equation gives

$$||\big(R(N, \bar{w}_\Lambda)\big)_{(I_j \setminus I'_j) \times J'_j}|| \leq \mathcal{O}(\sqrt{j \cdot \Lambda}). \tag{22}$$

In round $j = 1$, we start with $\ell_1 = k$ and $I_1 = J_1 = [k]$. For round $j > 1$ we chose, $\ell_j := \frac{\ell_{j-1}}{2}$, $I_j := I_{j-1} \setminus I'_{j-1}$ and $J_j := J_{j-1} \setminus J'_{j-1}$. Note that $I_j$ and $J_j$ are the excluded rows and columns in the concentration bounds of the previous round.

We use this procedure for $t = \lceil \log(k/\epsilon k) \rceil$ rounds, so that the weight and the number of excluded rows, and columns, in the end is at-most $\epsilon k$.

Let $\mathcal{M}_j := I_j \times J_j$, which is of size $k/2^{j-1} \times k/2^{j-1}$, $\mathcal{R}_j := I'_j \times J_j$ and $\mathcal{C}_j := I_j \setminus I'_j \times J'_j = I_{j+1} \times J'_j$. Note that equation 20 and 22 gives the concentration bound for sub-matrices corresponding to $\mathcal{R}_j$ and $\mathcal{C}_j$, respectively. Figure 1 shows this construction.

Figure 1: Construction of submatrices in proof of theorem 3.

This construction decomposes the submatrix indexed by $\mathcal{M}_j$ into three submatrices

$$\left(R(N, \bar{w}_\Lambda)\right)_{\mathcal{M}_j} = \left(R(N, \bar{w}_\Lambda)\right)_{\mathcal{R}_j} + \left(R(N, \bar{w}_\Lambda)\right)_{\mathcal{C}_j} + \left(R(N, \bar{w}_\Lambda)\right)_{\mathcal{M}_{j+1}}.$$

Applying the above equation recursively,

$$\begin{aligned}
\left(R(N, \bar{w}_\Lambda)\right) &= \left(R(N, \bar{w}_\Lambda)\right)_{\mathcal{M}_1} \\
&= \sum_{j=1}^t \left(R(N, \bar{w}_\Lambda)\right)_{\mathcal{R}_j} + \sum_{j=1}^t \left(R(N, \bar{w}_\Lambda)\right)_{\mathcal{C}_j} + \left(R(N, \bar{w}_\Lambda)\right)_{\mathcal{M}_{t+1}} \\
&= \left(R(N, \bar{w}_\Lambda)\right)_{\cup_{j=1}^t \mathcal{R}_j} + \left(R(N, \bar{w}_\Lambda)\right)_{\cup_{j=1}^t \mathcal{C}_j} + \left(R(N, \bar{w}_\Lambda)\right)_{\mathcal{M}_{t+1}},
\end{aligned}$$

where the last equality follows as $\mathcal{R}_j$'s and $\mathcal{C}_j$'s are disjoint. Then

$$\begin{aligned}
||\left(R(N, \bar{w}_\Lambda)\right)_{([k]\times[k])\setminus\mathcal{M}_{t+1}}|| &= ||R(N, \bar{w}_\Lambda) - \left(R(N, \bar{w}_\Lambda)\right)_{\mathcal{M}_{t+1}}|| \\
&= ||\left(R(N, \bar{w}_\Lambda)\right)_{\cup_{j=1}^t \mathcal{R}_j} + \left(R(N, \bar{w}_\Lambda)\right)_{\cup_{j=1}^t \mathcal{C}_j}|| \\
&\overset{(a)}{\le} ||\left(R(N, \bar{w}_\Lambda)\right)_{\cup_{j=1}^t \mathcal{R}_j}|| + ||\left(R(N, \bar{w}_\Lambda)\right)_{\cup_{j=1}^t \mathcal{C}_j}|| \\
&\overset{(b)}{\le} \sqrt{\sum_{j=1}^t \left(R(N, \bar{w}_\Lambda)\right)_{\mathcal{R}_j}} + \sqrt{\sum_{j=1}^t \left(R(N, \bar{w}_\Lambda)\right)_{\mathcal{C}_j}} \\
&\overset{(c)}{\le} 2 \cdot \mathcal{O}(\sqrt{\textstyle\sum_{j=1}^t j \cdot \Lambda}) \le \mathcal{O}(t\sqrt{\Lambda}) = \mathcal{O}(\log(k/\epsilon)\sqrt{\Lambda}),
\end{aligned}$$

where (a) follows from triangle inequality, (b) follows from Lemma 27, and (c) follows from inequalities (20) and (22). Note that $\sum_{i \in I_{t+1}} \bar{w}_\Lambda^f(i) = \sum_{i \in I_t \setminus I_t'} \bar{w}_\Lambda^f(i) \le k/2^t \le \epsilon k$ and, similarly, $\sum_{i \in J_{t+1}} \bar{w}_\Lambda^b(i) \le \epsilon k$.

Recall that $\mathcal{M}_{t+1} = I_{t+1} \times J_{t+1}$, therefore zeroing out rows $I_{t+1}$ from $([k] \times [k]) \setminus \mathcal{M}_{t+1}$ results in $([k] \setminus I_{t+1} \times [k])$. Since zeroing out rows of a matrix reduces the spectral norm, from the above equation we get

$$||\left(R(N, \bar{w}_\Lambda)\right)_{([k]\setminus I_{t+1}\times[k])}|| \le ||\left(R(N, \bar{w}_\Lambda)\right)_{([k]\times[k])\setminus\mathcal{M}_{t+1}}|| \le \mathcal{O}(\log(k/\epsilon)\sqrt{\Lambda}),$$

where $\sum_{i \in I_{t+1}} \bar{w}_\Lambda^f(i) = \sum_{i \in I_t \setminus I_t'} \bar{w}_\Lambda^f(i) \le k/2^t \le \epsilon k$. Letting $I_{t+1}$ to be the set of contaminated rows completes the proof of the theorem. ∎

In the next subsection, we derive a useful implication of Lemma 24.

### E.1  Proof of Lemma 20

The lemma upper bounds the sum of the absolute difference between the expected and observed samples in each row of $X$. We restate the lemma.

**Lemma.** *With probability* $\ge 1 - 3k^{-3}$,

$$\sum_i |\sum_j N_{i,j}| = \mathcal{O}(k\sqrt{n_{avg}}).$$

*Proof.* To prove the above Lemma we use Lemma 24, for $\Lambda = n_{\text{avg}}$, $\ell = k$ and $I = J = [k]$. The lemma implies that w.p. $\ge 1 - 3/k^3$,

$$\max_{v \in \{-1,1\}^k} ||D^{-\frac{1}{2}}(\bar{w}^f) \cdot N \cdot v|| = ||D^{-\frac{1}{2}}(\bar{w}^f) \cdot N||_{\infty \to 2} \le \mathcal{O}(\sqrt{n_{\text{avg}}k}).$$

From the definition of $\ell_\infty \to \ell_2$ norm,

$$\max_{v \in \{-1,1\}^k} ||D^{-\frac{1}{2}}(\bar{w}^f) \cdot N \cdot v|| = ||D^{-\frac{1}{2}}(\bar{w}^f) \cdot N||_{\infty \to 2}.$$

In the above equation choosing $v = (1, 1, ..., 1)$ and taking the square on the both side, we get

$$\sum_i \left(\sum_j \frac{N_{i,j}}{\sqrt{\bar{w}^f(i)}}\right)^2 \le ||D^{-\frac{1}{2}}(\bar{w}^f) \cdot N||_{\infty \to 2}^2 \le \mathcal{O}(n_{\text{avg}}k).$$

The above equation can be rewritten as,

$$\sum_i \frac{|\sum_j N_{i,j}|^2}{\bar{w}^f(i)} \leq \mathcal{O}(n_{\text{avg}}k).$$

Then

$$\sum_i |\sum_j N_{i,j}| = \sum_i \frac{|\sum_j N_{i,j}|}{\sqrt{\bar{w}^f(i)}} \cdot \sqrt{\bar{w}^f(i)} \leq \Big(\sum_i \frac{|\sum_j N_{i,j}|^2}{\bar{w}^f(i)}\Big)^{\frac{1}{2}} \cdot \Big(\sum_i \bar{w}^f(i)\Big)^{\frac{1}{2}} = \mathcal{O}(k\sqrt{n_{\text{avg}}}).$$

here we used the Cauchy-Schwarz inequality, the previous equation, and Lemma 18 that states $\sum_i \bar{w}^f(i) \leq 2k$. ∎

## F  Counterexample

The authors of [LLV17] posed the following question, whose affirmative answer may have simplified low-rank matrix recovery. They posed the question for Bernoulli-parameter matrix.

Let $M \in \mathbb{R}^{k \times k}$ be a Bernoulli-parameter matrix where $\forall i, j \in [k], ||M||_{i,*}, ||M||_{*,j} \leq n_{\text{max}}$, for some $n_{\text{max}}$.

Let $X = [X_{i,j}]$, where $X_{i,j} \sim \text{Ber}(M_{i,j})$, be the observation matrix of $M$, and let $X_0$ be the matrix obtained by zeroing-out the rows and columns of $X$ whose total count is $> 2n_{\text{max}}$. Does $X_0$ converges to $M$ w.h.p. as

$$||X_0 - M|| = \mathcal{O}(\sqrt{n_{\text{max}}})?$$

Unfortunately, the following counterexample answers this question in negative. Therefore additional work, such as presented in this paper, is needed to recover low-rank matrices.

For any $k$, choose any $n_{\text{max}} \leq \sqrt{\log k}/8$ that grows with $k$, and consider the block diagonal matrix

$$M = \begin{pmatrix} B_1 & & & & \\ & B_2 & & & \\ & & \ddots & & \\ & & & B_{(\frac{k}{2n_{\text{max}}}-1)} & \\ & & & & B_{\frac{k}{2n_{\text{max}}}} \end{pmatrix}$$

consisting of $k/(2n_{\text{max}})$ (for simplicity assume it is an integer) identical blocks $B_i = B$, each a submatrix of size $2n_{\text{max}} \times 2n_{\text{max}}$ whose entries are all $1/2$. Except for the blocks $B_i$'s, all the other entries of $M$ are zero. Then $M$ satisfies $\forall i, j \in [n], ||M||_{i,*} = ||M||_{*,j} = n_{\text{max}}$.

Note that, for the above matrix $n_{\text{avg}} = ||M||_1/k = n_{\text{max}}$.

The observation matrix of $M$ is

$$X = \begin{pmatrix} \hat{B}_1 & & & & \\ & \hat{B}_2 & & & \\ & & \ddots & & \\ & & & \hat{B}_{(\frac{k}{2n_{\text{max}}}-1)} & \\ & & & & \hat{B}_{\frac{k}{2n_{\text{max}}}} \end{pmatrix}.$$

Note that $X$ has non-zero entries only in locations corresponding to the diagonal blocks $B$. Also, $\forall i, j \in [k], ||X||_{i,*}, ||X||_{*,j} < 2n_{\text{max}}$. Therefore zeroing out rows and columns of $X$ with more than $n_{\text{max}}$ ones would not affect it and $X_0 = X$. From Theorem 6,

$$||X - M|| \geq \max_i ||\hat{B}_i - B_i||.$$

Since $\hat{B}_i$ is the observation matrix for the $2n_{\text{max}} \times 2n_{\text{max}}$ block $B_i$ whose entries are all $1/2$, and $n_{\text{max}} = o(\sqrt{\log k})$,

$$\Pr\left(\hat{B}_i = 0\right) = (1/2)^{4n_{\text{max}}^2} > 1/\sqrt{k}.$$

The probability that the whole $2n_{\max} \times 2n_{\max}$ block $\hat{B}_i$ is 0, is $(1/2)^{4n_{\max}^2}$, which since $n_{\max} = o(\sqrt{\log k})$, is $> 1/\sqrt{k}$. Hence w.h.p., at least one of the block $j \in [\frac{k}{2n_{\max}}]$ in $X$ is zero. Hence w.h.p.,

$$||X - M|| \ge ||B_j|| = n_{\max} \gg \Omega(\sqrt{n_{\max}}).$$

This counterexample answers the question raised by the authors of [LLV17] in negative.

We note that the same counterexample works for the regularization $R(X - M, \bar{w})$ used in this paper. Because if block $\hat{B}_j$ of $X$ is zero, from the definition of $\bar{w}$, it is easy to see that for the rows and columns corresponding to the block $\hat{B}_j$, the regularization weights are 1, which implies that

$$||R(X - M, \bar{w})|| \ge ||B_j|| = n_{\max},$$

extending the counterexample for the regularization $R(X - M, \bar{w})$ as well.

## G  Linear Algebra Proofs

### G.1  Proof of Lemma 4

**Lemma.** *For any rank-$r$ matrix $A \in \mathbb{R}_r^{k \times k}$, matrix $B \in \mathbb{R}^{k \times k}$, and weights $w$,*

$$||A - B^{(r,w)}||_1 \le \sqrt{r \cdot (\sum_i w^f(i))(\sum_j w^b(j))} \cdot ||R(A - B, w)||.$$

*Proof.* Recall that

$$B^{(r,w)} := D^{\frac{1}{2}}(w^f) \cdot R(B, w)^{(r)} \cdot D^{\frac{1}{2}}(w^b),$$

where $R(B, w) = D^{-\frac{1}{2}}(w^f) \cdot B \cdot D^{-\frac{1}{2}}(w^b)$ is regularized matrix $B$ and $R(B, w)^{(r)}$ is its rank $r$-truncated SVD.

We first upper bound the spectral norm of $R(A, w) - R(B, w)^{(r)}$ in terms of the spectral norm of $R(A - B, w)$. By Weyl's Inequality 7, and the rank $r$ of $A$,

$$\sigma_{r+1}(R(B, w)) \le \sigma_{r+1}(R(A, w)) + ||R(A, w) - R(B, w)|| = ||R(A - B, w)||.$$

Hence by the triangle inequality and the salient property of truncated SVD's,

$$||R(A, w) - R(B, w)^{(r)}|| \le ||R(B, w)^{(r)} - R(B, w)|| + ||R(A, w) - R(B, w)||$$
$$= \sigma_{r+1}(R(B, w)) + ||R(A - B, w)|| \le 2||R(A - B, w)||.$$

Since $A - B_r^{\text{SVD}}$ is the difference of two rank-$r$ matrices, it has rank $\le 2r$. Then applying Lemma 17 for matrix $(R(A, w) - R(B, w)^{(r)})$, and noting that $A = D^{\frac{1}{2}}(w^f) \cdot R(A, w) \cdot D^{\frac{1}{2}}(w^b)$ completes the proof. ∎

### G.2  Proof of Lemma 9

**Lemma.** *For any matrix $A$ and weight vectors $w^f$ and $w^b$ with positive entries*

$$||D^{-\frac{1}{2}}(w^f) \cdot A \cdot D^{-\frac{1}{2}}(w^b)|| \le \sqrt{\max_i \frac{||A_{i,*}||_1}{w^f(i)} \times \max_j \frac{||A_{*,j}||_1}{w^b(j)}}.$$

*Proof.* For a unit vector $v = (v(1), \ldots, v(m)) \in R^m$,

$$||D^{-\frac{1}{2}}(w^f) \cdot A \cdot D^{-\frac{1}{2}}(w^b) \cdot v||^2 = \sum_i \Big(\sum_j \frac{A_{i,j}v(j)}{\sqrt{w^f(i) \cdot w^b(j)}}\Big)^2$$

$$\le \sum_i \Big(\sum_j \frac{|A_{i,j}||v(j)|}{\sqrt{w^f(i) \cdot w^b(j)}}\Big)^2$$

$$= \sum_i \Big(\sum_j \sqrt{\frac{|A_{i,j}|}{w^f(i)}} \sqrt{\frac{|A_{i,j}|}{w^b(j)}} |v(j)|\Big)^2$$

$$\overset{(a)}{\leq} \sum_i \left( \left( \sum_j \frac{|A_{i,j}|}{w^f(i)} \right) \left( \sum_j \frac{|A_{i,j}|}{w^b(j)} v(j)^2 \right) \right) \leq \max_{i'} \frac{||A_{i',*}||_1}{w^f(i')} \sum_i \sum_j \left( \frac{|A_{i,j}|}{w^b(j)} v(j)^2 \right)$$

$$= \max_{i'} \frac{||A_{i',*}||_1}{w^f(i')} \sum_j \left( v(j)^2 \sum_i \frac{|A_{i,j}|}{w^b(j)} \right) \leq \max_{i'} \frac{||A_{i',*}||_1}{w^f(i')} \sum_j v(j)^2 \frac{||A_{*,j}||_1}{w^b(j)}$$

$$\leq \max_{i'} \frac{||A_{i',*}||_1}{w^f(i')} \times \max_{j'} \frac{||A_{*,j'}||_1}{w^b(j')} \sum_j v(j)^2 = \max_{i'} \frac{||A_{i',*}||_1}{w^f(i')} \times \max_{j'} \frac{||A_{*,j'}||_1}{w^b(j')},$$

where (a) uses the Cauchy-Schwarz inequality. Observing that above is true for arbitrary unit vector $v$ completes the proof. ∎

### G.3 Proof of Lemma 10

**Lemma.** *Let $A$ be an $k \times m$ matrix such that $\sigma_1(A) \leq \alpha$ and $\sigma_{r+1}(A) \leq \beta$. Then the number of disjoint row subsets $I \subset [k]$ such that $||A_I|| > 2\beta$ is at most $\left( \frac{r\alpha}{\beta} \right)^2$.*

*Proof.* Let $A = \sum_{i=1}^{\min\{k,m\}} \sigma_i(A) u_i v_i^\mathsf{T}$ be the SVD decomposition of $A$. Recall that $A^{(r)} = \sum_{i=1}^{r} \sigma_i(A) u_i v_i^\mathsf{T}$ and let $B = A - A^{(r)}$.
Note that $||A^{(r)}|| = \sigma_1(A) \leq \alpha$ and the matrix $A^{(r)}$ has rank $r$ i.e. $\sigma_{r+1}(A^{(r)}) = 0$. And $||B|| = \sigma_1(B) = \sigma_{r+1}(A) \leq \beta$.
To prove the lemma we upper bound the number of disjoint subsets $I \subset [k]$ such that $||A_I|| > 2\beta$.
Let $I \subset [k]$ be one such subset such that $||A_I|| > 2\beta$. Then

$$||A_I|| \overset{(a)}{\leq} ||A_I^{(r)}|| + ||B_I|| \overset{(b)}{\leq} ||A_I^{(r)}|| + ||B|| \leq ||A_I^{(r)}|| + \beta,$$

where inequality (a) follows from the triangle inequality and (b) follows from Theorem 6. Hence,

$$||A_I^{(r)}|| \geq \beta.$$

Note that since row span of $A^{(r)}$, and hence $A_I^{(r)}$ is $span\{v_1, v_2, ..., v_r\}$, therefore there exists a unit vector, $v = \sum_{i=1}^{r} a_i v_i$ (here $\sum_{i=1}^{r} a_i^2 = 1$, since $v$ is a unit vector), such that $||A_I^{(r)} v|| \geq \beta$. Therefore,

$$\beta \leq ||A_I^{(r)} v|| = ||A_I^{(r)} \sum_{i=1}^{r} a_i v_i|| \leq \sum_{i=1}^{r} |a_i| \, ||A_I^{(r)} v_i|| \leq \sum_{i=1}^{r} ||A_I^{(r)} v_i||. \tag{23}$$

Let $I_1, I_2, ...., I_t$ be the $t$ disjoint blocks such that $||A_{I_j}|| > 2\beta$, $\forall j \in [t]$. Next,

$$\sum_{i=1}^{r} ||A^{(r)} v_i|| \geq \sum_{i=1}^{r} ||A_{\cup_{j=1}^t I_j}^{(r)} v_i|| = \sum_{i=1}^{r} || \sum_{j=1}^{t} A_{I_j}^{(r)} v_i||$$

$$\overset{(a)}{=} \sum_{i=1}^{r} \sqrt{\sum_{j=1}^{t} ||A_{I_j}^{(r)} v_i||^2} \overset{(b)}{\geq} \sum_{i=1}^{r} \frac{\sum_{j=1}^{t} ||A_{I_j}^{(r)} v_i||}{\sqrt{t}} \overset{(c)}{\geq} \sqrt{t}\beta.$$

Here equality (a) follows since $A_{I_j}^{(r)} v_i$'s for $j \in [t]$ and fixed $i$ are orthogonal. Inequality (b) follows from the AM-GM inequality (c) follows from (23).
We also have $\sum_{i=1}^{r} ||A^{(r)} v_i|| = \sum_{i=1}^{r} \sigma_i(A) \leq r\sigma_1(A) \leq r\alpha$. Therefore we get, $t \leq \left( \frac{r\alpha}{\beta} \right)^2$. ∎

### G.4 Proof of Lemma 16

**Lemma.** *Let $A = B + C$ and $A = \sum_i \sigma_i(A) u_i v_i^\mathsf{T}$ be the SVD decomposition of $A$. And $\sigma_{r+1}(B) \leq \beta$ and $||C v_i|| \leq 2\beta$ for $i \in [2r]$. Then $\sigma_{2r}(A) \leq 4\beta$.*

*Proof.* Let $A^{(2r)} = \sum_{i=1}^{2r} \sigma_i(A) u_i v_i{}^{\mathsf{T}}$ be rank $2r$ truncated SVD of $A$, then

$$\sigma_i(A^{(2r)}) = \sigma_i(A), \; \forall \, i \leq 2r \tag{24}$$

and

$$A^{(2r)} = \sum_{i=1}^{2r} A v_i v_i{}^{\mathsf{T}} = \sum_{i=1}^{2r} (B+C) v_i v_i{}^{\mathsf{T}} = \hat{B} + \hat{C}. \tag{25}$$

Here $\hat{B} = \sum_{i=1}^{2r} B v_i v_i{}^{\mathsf{T}}$ and $\hat{C} = \sum_{i=1}^{2r} C v_i v_i{}^{\mathsf{T}}$.

Since $v_i$'s are orthogonal unit vector, $\sum_{i=1}^{2r} v_i v_i{}^{\mathsf{T}}$ is a projection matrix for subspace $S = \mathrm{span}\{v_1, v_2, .., v_{2r}\}$. And for any Projection matrix $P$ we have, $||Pu|| \leq ||u||$. Therefore,

$$||\hat{B}^{\mathsf{T}} u|| = ||(\sum_{i=1}^{2r} v_i v_i{}^{\mathsf{T}}) B^{\mathsf{T}} u|| \leq ||B^{\mathsf{T}} u||. \tag{26}$$

Next, using Courant-Fischer theorem, $\forall \, i \leq \min\{k, m\}$, there exists a subspace $S_i^*$ with dimension $dim(S_i^*) = i$, such that

$$
\begin{aligned}
\sigma_i(\hat{B}^{\mathsf{T}}) &= \min_{u \in S_i^*, ||u||=1} ||\hat{B}^{\mathsf{T}} u|| \\
&\overset{(a)}{\leq} \min_{u \in S_i^*, ||u||=1} ||B^{\mathsf{T}} u|| \\
&\leq \max_{S: dim(S)=i} \min_{u \in S, ||u||=1} ||B^{\mathsf{T}} u|| \\
&\overset{(b)}{=} \sigma_i(B^{\mathsf{T}}) = \sigma_i(B),
\end{aligned}
$$

where inequality (a) uses (26) and (b) again from Courant-Fischer theorem. Therefore,

$$\sigma_i(B) \geq \sigma_i(\hat{B}), \; \forall \, i \leq \min\{k, m\}. \tag{27}$$

Using (24), (25), Weyl's inequality 7 and (27):

$$\sigma_{2r}(A) = \sigma_{2r}(A^{(2r)}) \leq \sigma_{r+1}(\hat{B}) + \sigma_r(\hat{C}) \leq \sigma_{r+1}(B) + \sigma_r(\hat{C}) \leq \beta + \sigma_r(\hat{C}). \tag{28}$$

Now

$$\hat{C}\hat{C}^{\mathsf{T}} = \sum_{i=1}^{2r} (Cv_i)(Cv_i)^{\mathsf{T}} = \sum_{i=1}^{2r} \hat{u}_j \hat{u}_j{}^{\mathsf{T}}.$$

Here $\hat{u}_j = Cv_i$, hence $||\hat{u}_j|| \leq 2\beta$. Note that $\hat{C}\hat{C}^{\mathsf{T}}$ and $\hat{u}_j \hat{u}_j{}^{\mathsf{T}}$'s are Hermitian matrices. Let $\lambda_i(.)$ denotes the $i^{th}$ largest eigenvalue of the matrix. Then

$$\lambda_i(\hat{C}\hat{C}^{\mathsf{T}}) = \sigma_i^2(\hat{C}).$$

For rank-1 matrices $\hat{u}_j \hat{u}_j{}^{\mathsf{T}}$,

$$\lambda_1(\hat{u}_j \hat{u}_j{}^{\mathsf{T}}) = ||\hat{u}_j||^2 \leq 4\beta^2 \quad \text{and} \quad \lambda_i(\hat{u}_j \hat{u}_j{}^{\mathsf{T}}) = 0, \; \forall j \in [2r], \quad i \geq 2.$$

Then using Lidskii's theorem [Bha13], leads to

$$
\begin{aligned}
\sum_{i=1}^{r} \lambda_i \Big( \sum_{i=1}^{2r} \hat{u}_j \hat{u}_j{}^{\mathsf{T}} \Big) &\leq \sum_{i=1}^{r} \lambda_i \Big( \sum_{i=1}^{2r-1} \hat{u}_j \hat{u}_j{}^{\mathsf{T}} \Big) + \sum_i \lambda_i \big( \hat{u_{2r}} \hat{u_{2r}}{}^{\mathsf{T}} \big) \\
&\leq \sum_{i=1}^{r} \lambda_i \Big( \sum_{i=1}^{2r-1} \hat{u}_j \hat{u}_j{}^{\mathsf{T}} \Big) + 4\beta^2.
\end{aligned}
$$

By repeated application of Lidskii's theorem, we get

$$\sum_{i=1}^{r} \lambda_i \Big( \sum_{i=1}^{2r} \hat{u}_j \hat{u}_j{}^{\mathsf{T}} \Big) \leq 8r\beta^2.$$

Since $\lambda_i$'s are decreasing, it follows

$$r\lambda_r \Big( \sum_{i=1}^{2r} \hat{u}_j \hat{u}_j{}^{\mathsf{T}} \Big) \leq 8r\beta^2 \; \Rightarrow \lambda_r(\hat{C}\hat{C}^{\mathsf{T}}) = \sigma_r^2(\hat{C}) \leq 8\beta^2. \tag{29}$$

Combining (28) and (29) we get the statement of the lemma. ∎

### G.5 Proof of Lemma 17

**Lemma.** *For any rank-r matrix $A \in \mathbb{R}^{k \times m}$ and weight vectors $w^f$ and $w^b$ with non-negative entries*

$$||D^{\frac{1}{2}}(w^f) \cdot A \cdot D^{\frac{1}{2}}(w^b)||_1 \leq \sqrt{r(\sum_i w^f(i))(\sum_j w^b(j))} \cdot ||A||.$$

*Proof.*

$$
\begin{aligned}
||A||_1 &= \sum_i \sum_j \sqrt{w^f(i) \cdot w^b(j)}|A_{ij}| \\
&\overset{(a)}{\leq} \sum_i \sqrt{w^f(i)}\sqrt{(\sum_j A_{ij}^2)(\sum_j w^b(j))} \\
&= \sqrt{\sum_j w^b(j)}\sum_i |\sqrt{w^f(i)}|\sqrt{(\sum_j A_{ij}^2)} \\
&\overset{(b)}{\leq} \sqrt{\sum_j w^b(j)}\sqrt{(\sum_i w^f(i))(\sum_i \sum_j A_{ij}^2)}, \\
&\overset{(c)}{=} \sqrt{\sum_j w^b(j)}\sqrt{(\sum_i w^f(i))}||A_{I \times J}||_F \\
&\overset{(d)}{\leq} \sqrt{r(\sum_j w^b(j))(\sum_{i \in I} w^f(i))} \cdot ||A_{I \times J}||,
\end{aligned}
$$

where (a) and (b) follow from the Cauchy-Schwarz Inequality, (c) from the definition of the Frobenius norm

$$||A||_F := \sqrt{\sum_{i,j} A_{ij}^2},$$

and (d) as

$$||A||_F \leq \sqrt{\text{rank}(A)}||A||. \qquad \blacksquare$$

### G.6 Proof of Lemma 27

**Lemma.** *Let $A \in \mathbb{R}^{k \times m}$ and $I_1, I_2, I_3, ..., I_t$ be $t$ disjoint subsets of $[k]$ such that $\cup_{j=1}^t I_j = [k]$. Then $||A|| \leq \sqrt{\sum_{j=1}^t ||A_{I_j}||^2}$.*

*Proof.* Let $v \in \mathbb{R}^m$ be a unit vector. Then,

$$Av = \sum_{j=1}^t A_{I_j} v = \sum_{j=1}^t w_j.$$

Here $w_j = A_{I_j} v$. Since $I_j$'s are disjoint, $w_j$'s are orthogonal.

$$||Av|| = \sqrt{\sum_{j=1}^t ||w_j||^2} \leq \sqrt{\sum_{j=1}^t \max_{||v^*||=1} ||A_{I_j} v^*||^2} = \sqrt{\sum_{j=1}^t ||A_{I_j}||^2}.$$

Noting that the above bound holds for any unit vector $v \in \mathbb{R}^m$ completes the proof. $\blacksquare$