[Reviews · NeurIPS 2020]

Review 1

Summary and Contributions: This paper studies a generally formulated problem about recovering a low-rank matrix from samples. As a special case, the authors work captures the problem of leaning a k x k probability matrix M given access to samples from M, under the assumption that M is low-rank (rank r). The authors show how to solve this problem up to TV distance epsilon using kr/eps^2 * log(r/eps) samples, which is tight up to the log factor. Their result improves on a recent paper from ITCS 2018. Their algorithm also applies to a number of related sampling problems, which have applications in recommender systems, community detection, etc. The algorithm presented by the authors is based on a relatively elegant framework: the most direct approach to recover M would be to compute a truncated SVD of X. This approach can be shown to work when M’s rows/columns have roughly equal l1 norm, but falls apart when the matrix is non-uniform. A natural approach to deal with the issue is to first normalize rows/columns by multiplying X on the left and right by a diagonal reweighing matrix which down-weights heavy rows and columns, compute a truncated SVD, then recover M by reweighting the result. If weights could be chosen based on the l1 norm of columns/rows in M, this approach would immediately work, but unfortunately we do not have access to M. The authors main result is that it’s okay to use column/row norms from X, but with some added care. In particular, the SVD needs to be run on a version of X with a certain “bad” set of rows removed two obtain an accurate result. To find this set, the authors employ an iterative algorithm which uses an approximation algorithm for the 0-1 knapsack problem, as desired in Section 4.1.

Strengths: The problem studied is an extremely natural statistical problem, and the authors give an essential tight bound using a creative algorithm. Even if M only had r non-zero columns, kr/eps^2 samples would be required to learn the matrix, so the fact that the method handles general rank r matrices seems interesting to me. I am not super familiar with the prior work, so cannot say with absolute confidence how interesting this result will be to those working on similar problem, but it does give very concrete improvements on a number of prior papers.

Weaknesses: - The work is purely theoretical — it doesn’t implement the suggested method or related heuristics and show its value on a machine learning problem. I thought this was a shame because the method is *relatively* simple — granted the algorithm for finding a “bad” subset of rows/columns in quite involved, but maybe this part of the algorithm could be replaced with a natural heuristic? It would even have been interesting to show experiments where the simple reweighted truncated SVD fails to illustrate the algorithms value.

Correctness: I wasn’t able to check the entire lengthy appendix, but the details I went through seemed correct.

Clarity: I found the introduction a bit difficult to read. It started off with details a large number of related problems before we got a sense of what problem the authors would actually be solving. In particular, when reading I skipped from line 22 straight to 48, before returning later. I think a better structure would be to present the general problem (or maybe just the distribution matrix example) and then explain later how the approach extends to all of these different settings. I also found the technical overview a bit too informal at times to be useful, so had to mostly work off the appendix. I would have preferred to see more technical details upfront — e.g. to fully understand why truncated SVD works when ||X-M|| \leq sqrt{navg} or to fully understand why reweighted truncated SVD works when you have access to l1 norms of M’s rows + columns, which would have helped clarify why the subset removal approach is necessary.

Relation to Prior Work: Yes, very thoroughly discussed.

Reproducibility: Yes

Additional Feedback: Line 13: I found the intro a bit disorienting — I jumped into applications before we new what the basic problem setup is. Maybe a Moore direct approach would be to state the problem upfront, as in HKKV18, then go into applications. Line 2: as -> is Line 70: “is L1 distance, the standard Machine-Learning accuracy measure.” I think this could be worded better… it’s a common accuracy measure for comparing distributions, but for comparing matrices in ML L_2 is far more common. Line 76: It would be helpful to have some specific citations here. Line 84: What is the actual meaning of w_M? Giving an upper bound is a bit unsatisfying here: at least a quick comment on what the parameter is would be helpful. Line 99: What model is Theorem 2 being stated for? M is a matrix, and at this point I don’t think X \sim M has actually been defined… is X drawn from the distribution matrix model (which I think is the model from HKKV18)? I guess this result applies to all models? Line 125: Regularized in what way? Small = ? I found this statement too informal to be useful beyond what was stated in the previous paragraph. Line 202: What is the meaning for the sentence: “Note that the spectral distance between X and M is the same as the the spectral norm of the noise matrix N”? Does spectral distance between X and M have some meaning beyond ||N||? I didn’t see it defined elsewhere. Line 209: Where is this result from? Is it via a simple L1/L2 norm bound? — I guess it follows from specializing Lemma 17 to the case when the weights are 1 + a triangle inequality. I found this helpful to work through to get intuition — maybe there derivation could be included in the paper? Line 217: What is A? Is this the same as M? Line 233: Could you clarify this comment? Would an algorithm with say k*r*log(k)^c sample complexity be much more direct/easy to establish? Line 254: What does “contaminated” mean in this context? Line 309: So is the final estimate for m rank 2r instead of rank r? Is this a draw back? Is there a way to round to a rank r result?


Review 2

Summary and Contributions: This paper discussed the problem of estimating a low rank, real valued, k-by-k matrix M of rank r, from unbiased noisy samples X, i.e., where E(X) = M, under a (relative) L1 loss function. The authors demonstrate how this question underlies a multitude of well-known problems, some of which they discuss in some detail. They also show that the L1 norm ||M||_1 of the underlying matrix is proportional to the number of observations in these special cases. They then prove, based on known results in the minimax estimation literature, that ||M||_1 = \Omega(kr/eps^2) is required in order to achieve an \eps loss. As their main result, the provide an efficient algorithm that achieves the lower bound up to logarithmic terms in r and \eps. The algorithm, termed curated SVD, is a regularized variant of the standard truncated SVD, which essentially performs "outlier detection", seeking to eliminate (by zeroing-out) a small set of rows (if such exists) that has a disproportionate effect on the truncated SVD. The idea itself is reminiscent of standard techniques in regularized regression and outlier detection, yet the proof that this approach essentially achieves the lower bound appears to be novel.

Strengths: The motivation for the work is well established by many important examples of low-rank matrix recovery problems that fall under the setting. The suggested algorithm, while largely based on relatively standard ideas in regularized regression, is efficient and optimal up to log factors -- this appears to be a new result. though I must say that I am not sufficiently familiar with the literature on this topic to be certain. The technical part of the paper appears sound to the best of my limited reading, and the derivations are well explained.

Weaknesses: + The paper deals with estimating the low-rank mean of random matrices, and not in "learning low-rank distributions" as claimed in the title, which I found misleading. Only the mean of the distribution is being learned. + The abstract claims both an upper and a lower bound of the number of samples (L1 norm of M). However, the lower bound is well known in minimax estimation theory. I think this misleads the reader regarding the true contribution of the paper, which is an efficient almost-optimal algorithm. + In addition to the above, I suspect (but haven't verified) that it is also known (or not too difficult to prove) that the kr/eps^2 performance can be achieved in the minimax sense, without the additional log factors, by a non-efficient estimator (e.g., sieve maximum likelihood or similar). Perhaps this can even be deduced from [KOPS15], not sure. If I am correct, then the contribution of the paper is strictly limited to the efficiency of the algorithm. If not , then the contribution is bigger, proving that the lower bound is essentially tight. I urge the authors to address this and modify the paper accordingly. + The main result of the paper makes no assumption on the statistics on X, except that E(X) = M. But there must be some assumption on the conditional distribution p(X|M), right? It is not even stated whether this is an i.i.d. additive noise model, or a more general kernel. Even if it is i.i.d. additive, then there must be some assumption on the tail of the distribution of the noise X-M that affects the performance of the algorithm. I was not able to quickly locate where such an assumption is being used in the proof, but it must implicitly be there somewhere. This assumption and its effect on the performance should be clearly stated in Theorem 2.

Correctness: The claims appear to be sound, and the intuition is well explained.

Clarity: Absolutely.

Relation to Prior Work: Not sufficiently, as discussed above,

Reproducibility: Yes

Additional Feedback:


Review 3

Summary and Contributions: This paper studies the problem of recovering the entries of a latent low rank matrix M in R^{k x k}, where r < k is the rank. The matrix M is accessed through an observations of its entries generated by a random process. In most of the common modeling scenarios, X_{i,j} is the number of times a given pair (i,j) is observed; For instance, in latent semantic analysis, (i,j) is the probability of observing a (word,document) tuple, and in recommendation systems, X_{i,j} is the number of times customer i purchased product j. The authors present a unifying framework for modeling such latent matrix recovery problems. In this model, the latent matrix M is scaled to be the expectation of the random observations, namely M_{i,j} = E[X_{i,j}] for some random matrix X which is observed by the algorithm. The actual distribution of X_{i,j} can be drawn from several models, such as the Poisson, Bernoulli, and Binomial distributions. This framework captures several well-studied models, and in each model the number of total samples or observations used by the learning algorithm corresponds to the entry-wise L_1 norm of M. Namely, given a single sample X ~ M, the sample complexity is |M|_1. For example, in the recommendations system setting, |M|_1 is the expected number of times any customer purchases a product, which in turn is the sample complexity of the model. In this and related settings, the assumption that M is low rank is indeed often reasonable and popular to consider. Given the sample X, the algorithm must output a good approximation M^est to M, where the accuracy measure is the normalized entry-wise L_1 distance: L(M^est) = |M^est - M|_1/|M|_1 If M is a distribution matrix, namely non-negative and sum_{ij} M_{ij} = 1, and all draws have |X|_1=1, then this error measure is exactly the TV-distance between M^est and M. The authors argue that this, as well as the connection to sample complexity, makes L_1 particularly well suited for ML applications (as opposed to L_2, which is often more common in in most linear algebraic/low-rank problems). Their main results are nearly matching upper and lower bounds for the L_1 recovery error as a function of the L_1 norm of M (which again, corresponds to the sample complexity of the target applications). 1) Lower Bound: They demonstrate that there is a matrix M with |M|_1 < kr/eps^2 such any estimator of M via X ~ M has normalized L_1 expected error Omega(eps). 2) Upper Bound: They show a nearly matching upper bound, they give an algorithm with O(eps) error so long as |M|_1 > k r /eps^2 * log^2(r/eps) Thus, their upper bound matches the lower bound up to poly(log(r/eps)) factors.

Strengths: This results of this paper are quite nice, especially since the bounds are tight up to polylog factors. Moreover, the model is fairly simple and elegant (no extraneous assumptions are required), making it easy to compare to related work, and apply the recovery algorithms to existing problems. Recovery of low-rank matrices is an important and well-studied area, and I think this result would be interesting to anyone who works with learning latent models. The techniques are also interesting, and utilize non-trivial results from random matrix theory. Roughly, if the empirical error X-M converges has small spectral norm, as is the case for large classes of random matrices such as i.i.d. subgaussian entries, then recovery can be obtained by the (truncated) SVD. The main technical contribution of the authors is handling recovery when this is not the case, such as when a small subset of rows or columns have particularly large norm (i.e. large fraction of the L_1 mass hidden in a small subset of rows/columns). ----------------------- Post Rebuttal ----------------------- Thank you for clarifying the fact that multiple draws from the distribution are already built into the model. Moreover, I am happy to see this as a theoretical contribution, although an empirical evaluation would only make the paper stronger. I still feel favorably towards this paper, and will keep my original evaluation.

Weaknesses: While there is indeed good reason to study L_1, it would be interesting to understand the relation between the complexity of these problems also under L_2 and spectral norm bounds. This is not given much attention in the paper, although it is likely that some relevant work for both of these norms could be discussed at the very least. Also, it would be nice to see an empirical study of the proposed algorithm. The algorithm itself does not appear to be prohibitively complicated, so analyzing its performance in practice may be useful for the a NeurIPS Audience

Correctness: I did not fully verify the proofs, however the techniques and steps taken in the recovery algorithm and analysis are more than well-suited to solve the problem. It appears that all major steps taken in the algorithm and lower bound are correct.

Clarity: This paper is very well written. The informal discussion of the techniques, was very clear, and the proofs also are fairly easy to follow (relative to their technical depth).

Relation to Prior Work: A thorough comparison to prior work on the problem is given.

Reproducibility: Yes

Additional Feedback: It would be good to specify early on, perhaps even in the abstract, what exactly sample complexity means here. For instance, in some situations it may make sense to have multiple draws of X from the distribution specified by M. It may be interesting to understand the trade-offs in this case, ideally in the regime where the number of samples is small enough so that standard matrix concentration inequalities do not yet kick in. Regardless, perhaps mentioning this distinction could be useful, as it was not immediately obvious to me that one should not get multiple draws from M.


Review 4

Summary and Contributions: This paper provided theoretical analysis on learning low-rank matrices under the normalized L1 distance. More specifically, it showed that the theoretical sample complexity is linear in the high dimension, and also linear in the low rank. Moreover, the paper proposed a practical curated-svd algorithm to approximately match the theoretical bound, by first identifying and zeroing out a noisier subset of rows in the observation matrix. The paper is very novel in ideas and also practical in its implementation.

Strengths: The paper provided solid theoretical analysis on the low-rank matrix learning problem, and also proposed a practical algorithm (curated-svd) to implement the learning algorithm. The solution of this problem can cover a list of applications including collaborative filtering, community detection, etc. This paper would be an important contribution to our knowledge of this class of problem.

Weaknesses: It would be nice if the authors can also provide a section on numerical validation of the proposed curated-svd algorithm, for example, on simulated data and real-world datasets. The paper proposed to zeroing a subset of rows in the observation matrix to first reduce noise. Would doing that for rows and columns simultaneously could further help the performance and reduce the sample complexity? As for the low-rank matrix learning problem, the rows and columns are exchangeable, just by a transpose operation.

Correctness: As far as I can see, the theoretical analysis is correct, with details provided in the appendix. No empirical experiments are provided in the paper.

Clarity: Yes. The main body of the paper provided high-level conclusions, and the appendix provided a lot of technical details.

Relation to Prior Work: Yes. The paper provided good description of the related work. More specifically, the paper answered an open question that was raised in the literature [LLV17].

Reproducibility: Yes

Additional Feedback: It would make the paper more complete by adding a conclusion section, to summarize the full content of the paper, after "description of the Curated SVD". My assessment of the submission remains the same after viewing the authors' feedback.

[Author Response · NeurIPS 2020]

We thank the reviewers for the valuable time they have invested during this difficult period to review the paper and provide helpful suggestions for improving the manuscript. We also appreciate their complimentary comments and praise for the paper's theoretical contributions. In the following we address their comments, suggestions and questions.

Reviewers 1 and 2 ask about the precise relation between the parameter matrix $M$ and the observation matrix $X$, expressed by the notation $X \sim M$. As described in line 62 of section 1.2, the notation means that $M$ generates $X$ via one of the five models described in section 1.1. Later in the paper, we use this notation without reminding the reader of its meaning. Let us further clarify this notation here by briefly demonstrating it for two of the five models. In the paper's final version we similarly clarify the precise meaning of $X \sim M$, and repeat it before Theorem 2.

In the first model, *distribution matrices*, $M$ is a distribution matrix over $[k] \times [k]$ scaled by the number of times the distribution is sampled. Each time the distribution is sampled, a single independent value $(i, j)$ is observed, and $X$ is the total number of times element $(i, j) \in [k] \times [k]$ was observed among all samples taken. In the *Bernoulli matrices* model, $M$ is a matrix of Bernoulli parameters, and $X_{i,j} \sim \mathrm{Ber}(M_{i,j})$, independently, hence each $X_{i,j}$ is either 0 or 1.

The paper's unified formulation (lines 53-57) implies that in all 5 models (slight adjustment for #5), $X_{i,j}$ is the number of times $(i, j)$ is observed, and when $X \sim M$, $E[X] = M$. Hence the expected number of samples is $E[||X||_1] = ||M||_1$.

Reviewer 2 asks whether the paper's title is justified as we learn just the means of $X_{i,j}$, not their distribution. As we hope is now clear, we learn the parameter matrix $M$ that in all cases (except a small modification for collaborative filtering) determines the whole distribution of $X$. The fact that $E(X) = M$ is just a useful byproduct of our unified framework.

Reviewer 3 asks if multiple independent draws (of $X$) from the matrix $M$ may improve its estimate. That is correct. However, please note that such re-sampling is already accounted for in our models. For example, if $t$ i.i.d. instances $X^1, \ldots, X^t \sim M$ are generated using the Bernoulli model, the sufficient statistics would be the sum $X = \sum_i X^i$. This correspond exactly to $X \sim M' = tM$ using the Binomial model. A similar relation holds for all other models.

All four reviewers acknowledge the paper's technical contributions. However three reviewers point out that the algorithm is simple enough to implement and the experiments will improve the paper. Similar to the prior work that this paper improves on, e.g., [HKKV18] and [BCLS17], we viewed it mostly as a theoretical contribution, hence did not demonstrate its empirical efficacy. However, the algorithm is indeed simple to implement, and given the reviewers encouraging interest, we will gladly include simulations in the final version. From preliminary simulations, we expect good performance.

Reviewer 2 writes that the lower bound (Theorem 1) is well known, and should not be mentioned as a contribution. We are not aware however of such a lower bound for low-rank matrices, and if one is pointed out to us, we would gladly cite it. Yet the reviewer has a point in that the lower bound is very simple. We therefore don't even "prove" it, we just informally describe (lines 88-91) how it follows from previous work. We mention the lower bound to provide the reader a complete picture of the results and to show that the upper bound is essentially tight.

Reviewer 2 also asks whether the $kr/\epsilon^2$ upper bound for the non-efficient estimator's sample complexity exists in the literature, or may be easily obtained from [KOPS15], or from a sieve maximum likelihood estimator. We could not find any reference for the upper bound in the literature and would be happy to include one if pointed out. We also do not see how to obtain the result from [KOPS15], which does not leverage the structure of the distribution, hence will require $k^2$ samples, or from a sieve maximum likelihood estimator.

Reviewer 1 and 3 ask about learning in $L_2$ and spectral norms. Normalized $L_1$ norm is strictly stronger than normalized $L_2$ norm. Additionally, $L_1$ learning depends on the average row and column sums of $M$, yet as noted in [MD19] and mentioned in section 1.4, $L_2$ learning depends on the highest row and column sum, hence could suffer from a single heavy row. A similar observation about heavy rows and column sum holds for spectral norm as well. We will elaborate on this in the final version.

Reviewer 4 asks whether zeroing out rows and columns simultaneously could further reduce the algorithm's sample complexity. A small extension of Theorem 3 can show that difference matrix $X - M$ has a small submatrix (with few rows and columns) that when zeroed out will result in a remainder with a small spectral norm. Removing either the rows or the columns of this small submatrix would suffice, removing both would not yield significant additional advantage.

Reviewer 1 asks what are contaminated rows in Theorem 3. These are the rows we need to remove to achieve spectral concentration. Coincidentally, the same rows addressed in the previous paragraph. We will clarify that in the final version.

Reviewer 1 also asks whether statements in line 233 and 309 can be improved. Briefly, we believe the first can, but would require work. For the second, we do not know, but even if possible, it would improve the estimation error by at most a factor of two. We will also address typos and improve explanations as the reviewer suggests.

Finally, we would like to point out that although not addressed by the reviews, one consequence of our results, discussed in Section 1.4, significantly improves on state-of-the-art collaborative filtering results, e.g., [BCLS17].

[Meta-Review · NeurIPS 2020]

This paper studies a generally formulated problem about recovering a low-rank matrix from samples. As a special case, the authors work captures the problem of leaning a k x k probability matrix M given access to samples from M, under the assumption that M is low-rank (rank r). The authors show how to solve this problem up to TV distance epsilon using kr/eps^2 * log(r/eps) samples, which is tight up to the log factor improving on a recent ITCS paper. Their algorithm also applies to a number of related sampling problems, which have applications in recommender systems, community detection, etc. The reviewers were convinced of the contributions of this paper (though recommended adding experimental analysis of the algorithm on practical data). I am pleased to recommend accepting this paper to NeurIPS.